# Humans perseverate on punishment avoidance goals in multigoal reinforcement learning

Paul B Sharp[1,2,3]*[†], Evan M Russek[2,3][†], Quentin JM Huys[2,4], Raymond J Dolan[2,3], Eran Eldar[1]

[1]The Hebrew University of Jerusalem, Jerusalem, Israel; [2]Max Planck UCL Centre for Computational Psychiatry and Ageing Research, University College London, London, United Kingdom; [3]Wellcome Centre for Human Neuroimaging, University College London, London, United Kingdom; [4]Division of Psychiatry, University College London, London, United Kingdom

**Abstract** Managing multiple goals is essential to adaptation, yet we are only beginning to understand computations by which we navigate the resource demands entailed in so doing. Here, we sought to elucidate how humans balance reward seeking and punishment avoidance goals, and relate this to variation in its expression within anxious individuals. To do so, we developed a novel multigoal pursuit task that includes trial-specific instructed goals to either pursue reward (without risk of punishment) or avoid punishment (without the opportunity for reward). We constructed a computational model of multigoal pursuit to quantify the degree to which participants could disengage from the pursuit goals when instructed to, as well as devote less model-based resources toward goals that were less abundant. In general, participants (*n* = 192) were less flexible in avoiding punishment than in pursuing reward. Thus, when instructed to pursue reward, participants often persisted in avoiding features that had previously been associated with punishment, even though at decision time these features were unambiguously benign. In a similar vein, participants showed no significant downregulation of avoidance when punishment avoidance goals were less abundant in the task. Importantly, we show preliminary evidence that individuals with chronic worry may have difficulty disengaging from punishment avoidance when instructed to seek reward. Taken together, the findings demonstrate that people avoid punishment less flexibly than they pursue reward. Future studies should test in larger samples whether a difficulty to disengage from punishment avoidance contributes to chronic worry.

*For correspondence:
paul.sharp@mail.huji.ac.il

[†]These authors contributed equally to this work

Competing interest: The authors declare that no competing interests exist.

## Introduction

Adaptive behavior demands we flexibly shift between pursuit of multiple goals, but disengaging from one goal in order to pursue another is often challenging. Switching between different goals is computationally demanding as it requires us to disengage processing relevant to prior goals and recruit knowledge necessary to determine the best action to pursue new goals. Consider a teenager about to play for a championship of her basketball league, a coveted prize she is poised to attain. As the game begins, she suddenly remembers that earlier that day she again forgot to show up for a school exam, and consequently might end up getting expelled from school. Although current tasks demand she reallocate attention towards the basketball game, she persists in worry about a potential looming disaster awaiting when the game ends.

One possibility is that managing multiple goals is influenced by the valence of goal outcomes (i.e., goal valence) (*Guitart-Masip et al., 2012*). Thus, people might devote more resources to pursuing

goals involving potential punishment than to goals involving potential reward because of a tendency for losses to loom larger in magnitude than objectively equivalent gains (*Novemsky and Kahneman, 2018*). At the same time, people may adapt to their present environment such that a tendency to prioritize punishment avoidance might be attenuated if reward seeking goals are more frequently encountered than punishment avoidance goals. Thus, our first aim was to determine whether computational strategies for multigoal pursuit differ as a function of goal valence. Specifically, we investigated the degree to which individuals engage, and subsequently, disengage reward seeking and punishment avoidance goals under instruction, and how goal engagement and disengagement are impacted by the frequency with which the goals are encountered.

A striking example of a maladaptive preference for punishment avoidance manifests in individuals with pathological anxiety (*Bar-Haim et al., 2007*; *Berenbaum, 2010*; *Gagne and Dayan, 2021*; *Sharp and Eldar, 2019*; *Warren et al., 2021*). Such individuals tend to learn more quickly from punishment than reward (*Aylward et al., 2019*), and this can lead to avoidance of even moderately risky situations (*Charpentier et al., 2017*). Furthermore, evidence suggests that anxiety is associated with failing to terminate planning in relation to potential threats (*Berenbaum et al., 2018*; *Hunter et al., 2022*). However, anxiety-associated failures to effectively disengage punishment avoidance goals have not been examined in a task that tests people's ability to engage or disengage from punishment avoidance goals at will. Such a test is required to disambiguate between underlying computational mechanisms explaining how these failures occur (*Browning et al., 2015*; *Korn and Bach, 2019*).

On the one hand, it is possible that in naturalistic settings anxious individuals allocate more resources toward punishment avoidance because they believe the environment demands it, and thus, if given explicit safety signals they would effectively disengage punishment avoidance, perhaps even more so than less anxious individuals (*Wise and Dolan, 2020*). On the other hand, anxious individuals might fail to disengage punishment avoidance even in the presence of explicit safety signals, evincing a more fundamental failure in exercising executive control. Importantly, both hypotheses are consistent with anxious individuals opting for avoidance behavior in approach–avoidance conflict tasks (*Loh et al., 2017*), but diverge in settings where punishment avoidance and reward seeking goals are unambiguously separated in time and space. Thus, our second aim was to explore potential computations involved in disengagement of punishment avoidance goals in anxiety.

We developed a novel multigoal pursuit task that required participants to learn by trial and error the probabilities that different actions lead to different task features. Learning was incentivized by occasionally coupling certain features with monetary punishment and other features with monetary reward (*Figure 1*). Critically, on each trial, participants were instructed either to avoid the punishment feature or to seek the reward feature, and these goals switched frequently, requiring participants to continuously adjust their behavioral policy. Unbeknownst to participants, we manipulated how frequently certain goals were encountered in each task block, allowing us to determine whether more costly decision-making resources are devoted to pursuing more frequent, and thus more reward-impacting, goals in a resource rational (RR) manner (*Lieder and Griffiths, 2019*).

We report evidence that participants relied to varying degrees on three strategies. Whereas a model-based (MB) strategy was employed to learn the probabilities by which actions led to features for the purpose of flexibly pursuing instructed goals, there was also evidence for a model-free strategy that disregarded instructed goals and relied on points won or lost to reinforce actions (*Lai and Gershman, 2021*). Most interestingly, we find evidence for use of a novel strategy we term, 'goal perseveration' (GP), whereby participants learn feature probabilities akin to an MB strategy but utilize this knowledge in a less costly and less flexible way, so as to always avoid punishment (even when instructed to seek reward) and to always seek reward (even when instructed to avoid punishment). Strikingly, this GP strategy was used to a greater extent for punishment avoidance, suggesting that disengaging punishment avoidance is harder, perhaps for evolutionarily sensible reasons (*Woody and Szechtman, 2011*). By contrast, the more flexible MB strategy was leveraged to a greater degree during reward seeking. Furthermore, participants flexibly increased MB control toward reward seeking goals when they were more abundant.

Finally, in a series of exploratory analyses, we sought to determine whether and how anxious individuals express a preference for punishment avoidance goals. In so doing, we found preliminary evidence that the degree of reliance on a GP strategy to avoid punishment was positively associated

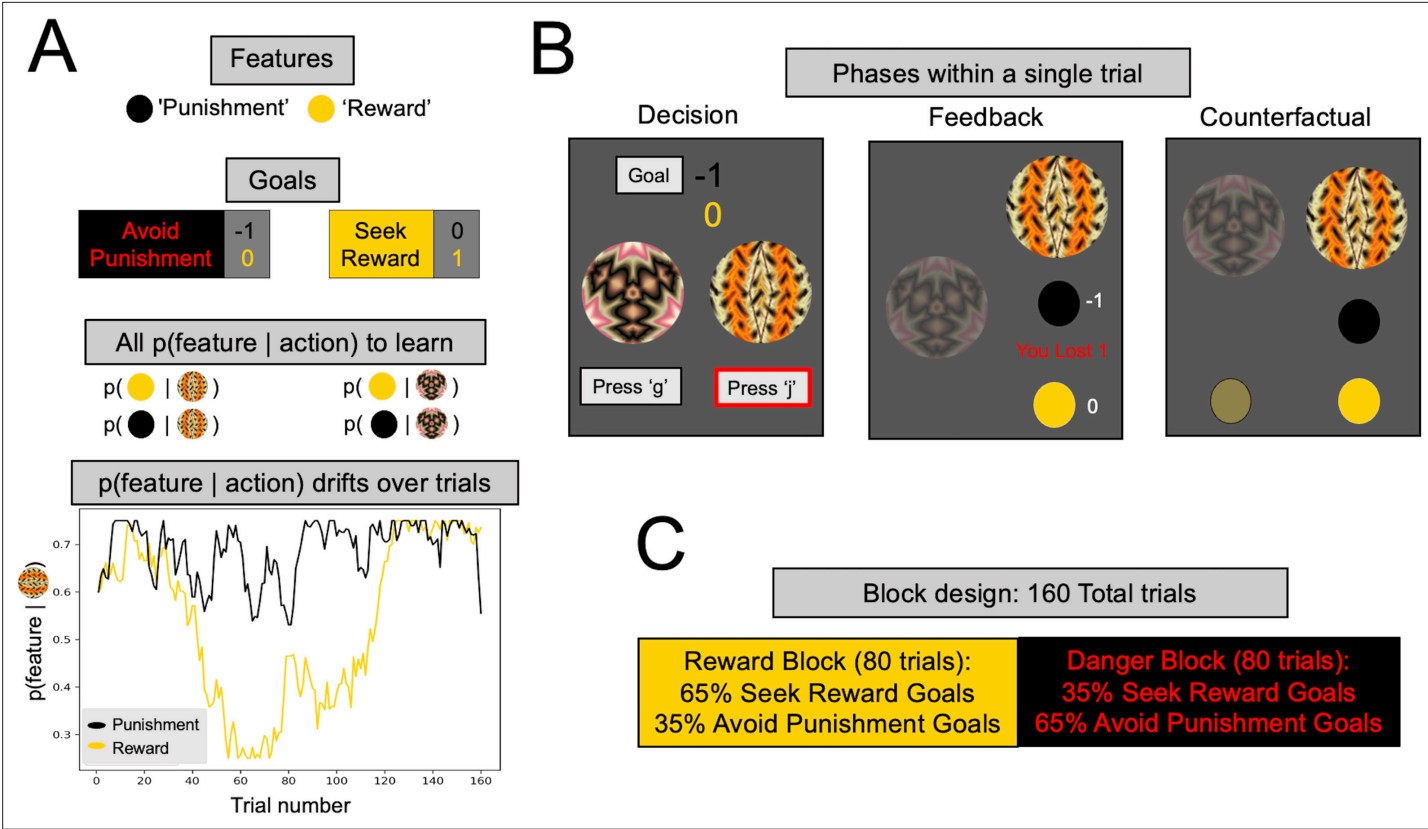

**Figure 1.** Multigoal pursuit task. (**A**) Key task components. Participants were instructed to learn the likelihood of observing two features (gold and black circles) after taking each of two actions (pressing 'g' or 'j' on the keyboard), and integrate this knowledge with instructed trial-specific goals denoting the present reward or punishment value of each feature. There were two possible goals: in one participants were instructed to seek the reward feature (reward feature = +1 point, punishment feature = 0) and in the other to avoid the punishment feature (reward feature = 0, punishment feature = −1 point). Thus, if the goal was to seek reward, participants should have selected the action most likely to lead to the reward feature (gold circle), irrespective of whether the action lead to the punishment feature (as the value of the latter is 0). Critically, whether each of the two features was present was determined independently, that is, for each action there were four different possible outcome configurations (both features present/reward feature only/punishment feature only/both features absent). To pursue goals, participants had to learn via experience four probabilities (left panel, all p(feature|action)) comprising the likelihood of observing each feature following each action (i.e., they were never instructed about these probabilities). Continued learning was required because the true probabilities of observing features for different actions drifted across trials according to semi-independent random walks (bottom left). Although participants were instructed with a more neutral narrative (see Methods), here we refer to the gold circle as the reward feature and the black circle as the punishment feature. However, the gold circle was rewarding *only* for reward seeking goal trials (and of no value during punishment avoidance goal trials), whereas the black circle was punishing *only* during punishment avoidance goal trials (and of no value during reward seeking goal trials). In the actual task implementation, the color for the reward and punishment features, and the random walks each feature took, were counterbalanced across participants. (**B**) Phases of a single trial. First, participants were shown both fractals and the current goal, and asked to select an action ('Decision'). After they took an action (here, clicking 'j', denoted by the red outline), participants were shown feedback, which comprised the feature outcomes, the present reward value of each feature, and the total points gained (possible total points were: (1) 'You lost 1', (2) 0, or (3) 'You won 1'). Finally, participants were shown the feature outcomes they would have seen had they chosen the other action ('Counterfactual'), which could be any of four possible feature combinations. (**C**) Goal abundance manipulation. A totality of 160 trials were divided into two equal length blocks, constituting reward- and punishment-rich contexts. In a reward-rich context, reward seeking trials outnumbered punishment avoidance trials, and the converse was true in a punishment-rich context. Note, both the sequence and order of blocks were counterbalanced across goal types to ensure neither factor could account for prioritizing a specific goal.

with dispositional worry, which appears to be unique to those expressing worry and not to individuals with obsessive–compulsive (OC) or somatic anxiety symptoms.

## Results
### Task description
We recruited a large online sample of participants ($N$ = 192, excluding 56 who did not meet performance criteria; excluded participants did not differ significantly on any psychopathology measure from the retained sample; see Methods) to play an online version of a novel multigoal pursuit task (**Figure 1**). At each trial, participants could take two possible actions, defined by fractal images, to seek or avoid certain features. The trial's goal was defined by the effective reinforcement value of the features, which varied from trial to trial as instructed to participants explicitly at the beginning of each trial. Thus, in reward seeking trials, encountering a 'reward' feature (gold circle) gifted the participant one point whereas the 'punishment' feature (black circle) was inconsequential (value = 0). By contrast, in punishment avoidance trials, the punishment feature took away one point whereas the reward feature had no value. Note that the reward value of either feature was continuously presented throughout the choice deliberation time (**Figure 1B**), ensuring that there should be no reason for participants to forget the present trial's goal. To determine whether participants adapted their decision-making toward more frequently encountered goals, we designed our task such that one goal was more abundant in each half of the task.

After participants made a decision, they were shown the choice outcome feature, followed by a counterfactual feature outcome for the choice not made (**Figure 1B**). The probabilities linking features to actions varied over time, and participants could estimate these continuously drifting probabilities from experience by observing which features actions led to. Presenting both actual and counterfactual outcomes removed the need for participants to explore less-visited actions to gain information, thus ruling out information seeking as a normative explanation for deviations from optimal choice. Of note, this task design differs from influential two-factor learning paradigms (**Mowrer, 1951**) extensively used to study anxiety, in that in our task both action-feature and feature-value associations changed throughout the experiment, mandating continued learning and flexible decision-making.

### Three computational strategies
#### Model based
We sought to identify computations individuals employed to learn and enact decisions in our task. A suitable computational strategy for this task is to continuously learn which task features follow each action irrespective of the instructed goal, and when deciding which action to take, rely specifically on knowledge about features relevant to the presently instructed goal. This strategy is an instance of a broader family of 'model-based' strategies that flexibly use knowledge about which actions lead to which states (**Dolan and Dayan, 2013**). By simulating an artificial MB agent, we show that a unique signature of MB control in our task manifests, when current and previous goals differ (henceforth referred to as 'goal-switch' trials), in the way the current goal determines whether observed features in the last trial impact subsequent action. For example, an MB agent will avoid an action leading to a punishment feature in the last trial only when the current instructed goal is to avoid punishment (**Figure 2A**, top row). Such behavior cannot be produced by the other strategies discussed subsequently unless the current and previous goals are the same.

#### Model-free
An MB strategy can be highly effective in our task, but it demands constant adaptation of one's actions to frequently changing goals. Thus, we expect participants might resort to less costly, approximate strategies (i.e., heuristics). One common heuristic simplifies MB strategies by learning which actions have higher expected value purely based on experienced rewards and punishments. This so-called 'model-free' (MF) reinforcement learning strategy is ubiquitously deployed in single-goal tasks (**Daw et al., 2011**; **Sutton and Barto, 2018**). In the present multitask setting, this would entail forgoing adaptation to the current goal and instead simply learning the overall expected values of the two available actions. Since the previous goal is what determines the value of the last observed features, a unique signature of an MF strategy is how a previous goal determines the impact of last

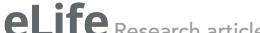

**Figure 2.** Behavioral signatures of computational strategies in simulated and real data. (**A**) Last outcome effects in simulated data. Each row comprises data generated by simulating one of the candidate computational strategies used to enact decisions in the present task (see Methods for parameters used). Each plot depicts the proportion of times the simulated agent takes a different action than that taken on the last trial ('switch probability'), as a function

*Figure 2 continued on next page*

*Figure 2 continued*

of features experienced on the last trial *for the chosen action* (gold/black circles; a gray bar indicates the feature was absent), the previous goal (left vs. right plots), and the current goal (light vs. dark bars). (**B**) Last outcome effects in empirical data. Real participants' switch probabilities as a function of last trial's feature outcomes, and current and previous goals. For illustration, we overlay repeated measures *t*-tests of specific MB (difference between blue and black bars) and GP (green bars) predictions, broken down by goal valence. A more thorough analysis of strategies used by participants is shown in panel C. *p < 0.05, **p < 0.01, *****p < 10⁻⁵, *******p < 10⁻⁷. (**C**) Empirical evidence for each strategy. Posterior distributions derived from fitting a Bayesian linear mixed-effects logistic regression evince main effects for MB (blue), GP (green), and MF (red) strategies. Evidence reflects MB and MF were leveraged for punishment avoidance and reward seeking goals whereas GP was leveraged for punishment avoidance goals, with only trending evidence it was used for reward seeking. (**D**) Effect of goal valence on strategy utilization. We estimated goal valence effects by examining the posterior distribution of differences between the parameters in panel C and found evidence indicating model-based utilization was greater for reward seeking, whereas goal-perseveration utilization was greater for punishment avoidance.

The online version of this article includes the following figure supplement(s) for figure 2:

**Figure supplement 1.** Resource rational simulations.

**Figure supplement 2.** Priors for group-level means and variances in our mixed-effects modeling for empirical data.

**Figure supplement 3.** Postpredictive check.

observed features on subsequent action, regardless of whether the goal has switched. For example, if an action led to a punishment on the last trial, then that action will tend to be avoided irrespective of the current goal (*Figure 2A*, bottom row).

## Goal perseveration

A MF strategy is relatively simple to implement but not particularly effective since it does not utilize the information provided by feature outcomes that currently have no reward or punishment value (i.e., a feature that is irrelevant given the trial's goal or that is a counterfactual outcome of the unchosen action). An alternative strategy, that we term 'goal perseveration', might strike a better balance between simplicity and effectiveness. This strategy inherits the exact same knowledge of feature probabilities acquired by MB learning, but simplifies action selection by persistently avoiding punishment and seeking reward, simultaneously, regardless of instructed goal. This, in principle, eliminates effortful goal switching while utilizing all available information about the changing action-feature mapping. Thus, rather than constituting a separate decision system in its own right, GP is best thought of as a behavior produced by a strategic cost-saving MB agent. In goal-switch trials, a GP strategy would manifest in the observed features having the same impact on subsequent actions regardless of the current or previous trial's instructed goal. For example, a GP agent will avoid an action that led to a punishment feature at the last trial even if both previous and current goals were to seek reward (*Figure 2A*, middle row).

## The benefits and costs of each strategy

MB strategies typically harvest more reward than heuristic strategies but are computationally costly, hence individuals will tend to use them to a greater extent when properly incentivized (*Konovalov and Krajbich, 2020*; *Kool et al., 2017*; *Patzelt et al., 2019*). To determine whether our task properly incentivized the use of an MB strategy, we simulated agents playing the task many times and computed the average amount of reward earned and punishment avoided with each computational strategy. This showed that an MB strategy in our task led to significantly more reward than the other strategies (*Figure 3A*; e.g., around 40% more than a GP agent), and only 15% worse than an idealized model that has access to the true feature probabilities for each action. The advantage of the MB strategy was due in large part to the task involving frequent goal switching (41.8% of trials). Finally, the least costly MF strategy also earns the least reward in the present task (*Figure 3A*).

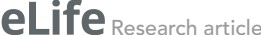

**Figure 3.** Task performance of distinct strategies. (**A**) Average total points gained by computational strategies. Punishment, reward, and total points (i.e., reward minus punishment) were averaged over 2000 simulations for each strategy. Strategies included model based (MB), model free (MF), and three versions of a goal perseveration (GP reward seeking with MB punishment avoidance [GP-R], GP punishment avoidance with MB reward seeking [GP-P], and GP for both reward and punishment goals [GP]). Details of parameters and models for each agent simulated are detailed in Methods. Each agent played the task 2000 times. Measures are range normalized such that 0 corresponds to performance of an agent that purely guesses and one corresponds to performance of the best-performing agent. (**B**) Punishment avoided by computational strategies. Here, the plot tallies successful attempts by agents to avoid punishment. The results illustrate that a hybrid agent that employs the goal-perseveration punishment avoidance strategy, and utilizes model-based control for reward seeking, avoids punishment as successfully as a fully model-based agent. (**C**) Reward earned by computational strategies. Here, the plot tallies successful attempts by agents to seek reward. This highlights that a hybrid agent that employs the goal-perseveration punishment avoidance strategy gains less reward than a model-based agent.

## Empirical evidence of each computational strategy

### Evidence of MB learning

To estimate whether participants leveraged each of the three strategies, we fit a Bayesian linear mixed-effects logistic regression to participant choices on goal-switch trials, wherein unique signatures of each strategy are detectable. Besides accounting for each strategy's signature, the regression controlled for the main effect of goal. The MB regression parameter predicted whether a participant switched to a new action on the current trial as a function of the interaction between the features observed last trial for chosen and unchosen actions and the instructed goal on the current trial (see Methods). Thus, we found a strong main effect of MB behavior (MB main effect mode = 0.59, confidence interval [CI] = [0.43,0.76], pd = 1; *Figure 2*).

Examination of the data prior to regression analysis suggested a difference in utilization of MB control for reward seeking relative to punishment avoidance (*Figure 2B*). To determine whether an MB effect was present for both punishment avoidance and reward seeking goals, we enhanced the regression with separate MB parameters for the two goals. Posterior estimates showed that individuals engaged MB control for both reward seeking (mode = −0.52, CI = [−0.69,−0.38], pd = 1) and punishment avoidance goals was highly trending (mode = 0.12, CI = [−0.01,0.28], pd = 0.96). Moreover, we found a larger MB effect for reward seeking than for punishment avoidance (mode = 0.41, CI = [0.20,0.62], pd = 1).

### Evidence of MF learning

We next determined whether participants also used an MF strategy, as captured by a regression parameter reflecting an interaction between the features observed at the last trial for chosen actions and the instructed goal on the last trial (this interaction represents the reward or punishment incurred last trial). Posterior estimates showed a MF strategy was employed by participants (MF main effect mode = 0.24, CI = 0.14,0.36, pd = 1), both in response to reward (mode = −0.14, CI = [−0.23,−0.04]; *Figure 2C*, bottom row) and punishment (mode = 0.22, CI = [0.12,0.31]). We found no evidence that the valence of the feedback impacted MF behavior to a greater degree (mode = −0.08, CI = [−0.21,0.06], pd = 0.87).

### Evidence of a GP strategy

Finally, we determined whether participants used a GP strategy, as captured by a regression parameter reflecting effects of reward and punishment features observed last trial irrespective of goal. We observed a strong GP effect (GP main effect mode = 0.26, CI = [0.14,0.40], pd = 1). Breaking the GP effect down by valence showed that GP was utilized for punishment avoidance (mode = 0.33, CI = [0.20,0.45], pd = 1), significantly more so than for reward seeking (mode = −0.11, CI = [−0.23,0.02], pd = 0.95; difference between goals: mode = −0.20, CI = [−0.37,−0.04], pd = 1).

## Quantifying the contribution of each strategy to decision-making

The presence of unique signatures of MB, MF, and GP decision strategies in the empirical data presents strong evidence for the use of these strategies, but the signature measures are limited to examining goal-switch trials and, within those trials, examining the impact of features observed on the very last trial. To comprehensively quantify the extent to which participants utilized each strategy for reward seeking and punishment avoidance, we next developed a series of computational models that aim to explain all participant choices given the features observed on all preceding trials.

We first sought to determine whether each strategy explained unique variance in participants' choices. To do so, we implemented a stepwise model comparison (see Methods for full details of the models) that began with a null model comprising only action perseveration (AP). Specifically, an AP strategy reflects the tendency of participants to stay with the action taken at the last trial, which has been found in various prior studies on single-goal reinforcement learning (*Daw et al., 2011*; Bayesian information criterion [BIC] = 38,273.53).

We subsequently investigated whether an MB strategy explained unique variance in participant's choices above the null. To do so, we compared the null model to a similar model where we added an MB strategy. We found that the MB model explained significantly more variance than the null model (Δ iBIC = −3900.94), a finding that coheres with our expectation that participants would utilize an MB strategy to make choices.

Before considering additional strategies, we asked whether individuals adjusted how they utilized an MB strategy for reward seeking and punishment avoidance in a 'resource rational' (RR) fashion, based on how abundant each goal was in the task block (MB$_{RR}$; *Figure 2—figure supplement 1*). Allowing the model to adjust in this way improved an index of model fit significantly (Δ iBIC = −175.95; *Figure 4A*) providing evidence that individuals reallocated MB resources toward goals that were more abundant.

We next tested whether a MF strategy explains unique variance in choice data beyond the MB$_{RR}$ model. To do so, we compared the MB$_{RR}$ model to a similar model that combined both MB$_{RR}$ and MF strategies. In controlling choice, the two strategies were combined via a weighted sum of the values they each assigned to a given action. Thus, a strategy's weight quantifies the degree to which it was

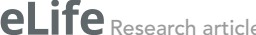

**Figure 4.** Results from computational modeling. *p < 0.05, ***p < 0.001. (**A**) Stepwise model comparison. The plot compares all models to the winning model 'MB$_{RR}$+ GP + MF'. (**B**) Model-based utilization is greater for reward seeking than for punishment avoidance. Here and in panel C, distributions are compared in terms of their medians due to a heavy positive skew. (**C**) Goal-perseveration utilization is greater for punishment avoidance than for reward seeking. Panels B and C show the distributions of utilization weights that best fitted each individual participant's choices.

The online version of this article includes the following figure supplement(s) for figure 4:

**Figure supplement 1.** Plots of symptom distributions and noting cutoff for clinical symptoms and percent of clinical symptoms.

**Figure supplement 2.** Parameter recovery.

**Figure supplement 3.** Alternative model descriptions and full model comparison.

**Figure supplement 4.** Correlations between fitted parameters.

utilized. The MB$_{RR}$+ MF models' fit was superior to the MB$_{RR}$ model (Δ iBIC = −896.72; *Figure 4A*), providing evidence that individuals used both MF and MB$_{RR}$ strategies to inform their decisions.

Finally, we tested whether a GP strategy might explain additional variance beyond the MB$_{RR}$ and MF strategies. Enhancing the MB$_{RR}$ + MF with the addition of a GP strategy significantly improved model fit (Δ iBIC = −243.73; *Figure 4A*), indicating participants also used a GP strategy. Henceforth, we refer to this final MB$_{RR}$ + MF + GP model as the 'winning model' (see Methods for full model formalism). To further validate this winning model, we compared it to several alternative models that were found inferior in fitting the data, including models where we removed GP, MF, and MB$_{RR}$ processes to ensure the order in which we added each strategy did not impact the final result, and an alternative MF account wherein the goal was encoded as part of state representation (see *Figure 4—figure supplement 3* for model specifications and model comparison results). Ultimately, we showed

that generating data from the winning model, using best-fitting participant parameters, could account for all but one of the mechanism-agnostic results reported in *Figure 2C and D* (*Figure 2—figure supplement 3*).

## Punishment avoidance is less flexible than reward seeking

To investigate algorithmic differences between reward seeking and punishment avoidance, we used the winning model to extract the parameter values that best fit each participant. We focused our analysis on parameters quantifying the degree to which individuals utilized a particular strategy to pursue reward or avoid punishment goals. We validated that MB, GP, and MF inverse temperature parameters were recoverable from simulated experimental data, and that the degree of recoverability (i.e., the correlations of true and recovered parameter values, which were between 0.76 and 0.91; *Figure 4—figure supplement 2*) was in line with extant reinforcement learning modeling studies (*Palminteri et al., 2017*; *Haines et al., 2020*). Similarly, low correlations between estimated parameters (all weaker than 0.16) demonstrate our experimental design and model-fitting procedure successfully dissociated between model parameters (*Wilson and Collins, 2019*).

In doing so, we found that individuals relied significantly more on an MB strategy for reward seeking relative to punishment avoidance (two-tailed p < 0.001, nonparametric test [see Methods]; *Figure 4B*). By contrast, individuals relied more heavily on GP for punishment avoidance relative to reward seeking (two-tailed p = 0.026, nonparametric test; *Figure 4C*). These results suggest participants did not adaptively 'turn off' the goal to avoid punishment to the same extent as they did so for the goal to pursue reward.

Finally, we examined whether individuals prioritized punishment avoidance and reward seeking goals based on their relative abundance. To do so, we extracted computational parameters controlling a shift in MB utilization across task blocks for both goal types. Each of these utilization change parameters was compared to a null value of 0 using a nonparametric permutation test to derive valid p values (see Methods). This analysis revealed that individuals were sensitive to reward goal abundance (mean = 0.50, p = <0.001) but not to punishment goal abundance (mean = −0.13, p = 0.13). This result comports with previous results which highlighted a difficulty disengaging from punishment avoidance. Moreover, this result points to why our winning model, that allowed MB utilization weights to change across task block, explained participant data better than a model that kept MB utilization weights constant.

## Preliminary evidence chronic worry is associated with greater perseverance of punishment avoidance

In a set of exploratory analyses, we sought to investigate how anxiety might be related to a prioritization of punishment avoidance goals. To do so, we assayed participants using self-report measures of chronic worry (*Meyer et al., 1990*) and somatic anxiety (*Casillas and Clark, 2000*) and OC symptoms (*Figure 4—figure supplement 1*). For each regression model, we computed p values using a nonparametric permutation test wherein we shuffled the task data with respect to the psychopathology scores, repeating the analysis on each of 10,000 shuffled datasets to derive an empirical null distribution of the relevant *t*-statistics.

We first report the bivariate relations between each form of psychopathology and inverse temperature parameters reflecting tendencies to utilize MB and GP punishment avoidance. Given that individuals with OCD and anxiety symptoms may overprioritize threat detection, it is conceivable that there is a relationship between all three forms of psychopathology and MB punishment avoidance. However, we found no significant or trending relationships between any form of psychopathology and MB control for punishment avoidance (*Figure 5A*, left column). An alternative possibility is that individuals with anxiety suffer from a dysregulation in goal pursuit, reflecting a failure to disengage punishment avoidance when instructed to do so. On this basis, we explored whether worry and somatic anxiety are positively associated with GP for punishment avoidance. In so doing we found initial evidence of a potential relationship between the tendency to worry and punishment avoidance perseveration (*B* = 2.15, *t* = 1.4, p = 0.16; *Figure 5A*, right column).

To provide a more specific test of our key hypotheses, we removed variance of noninterest in order to sensitize our analyses to the unique relationships between forms of psychopathology and types of punishment avoidance. Firstly, generalized, as opposed specific obsessive, worry is thought to be

|   | MB Punish | GP Punish |
|---|---|---|
| **A** | | |
| Worry | -1.32 (1.27), p=0.3 | 2.15 (1.53), p=0.16 |
| OCD | 0.99 (1.01), p=0.33 | -0.85 (1.22), p=0.49 |
| Somatic Anxiety | 0.4 (0.47), p=0.4 | -0.09 (0.57), p=0.88 |

|   | MB Punish | GP Punish |
|---|---|---|
| **B** | | |
| Worry | -1.39 (1.2), p=0.25 | 3.14 (1.38), p=0.02 |
| OCD | 1.11 (0.93), p=0.24 | -1.85 (1.08), p=0.09 |
| Somatic Anxiety | 0.52 (0.48), p=0.28 | -0.02 (0.56), p=0.97 |

**Figure 5.** Exploratory relationships between threat-related psychopathology and goal-directed control for punishment avoidance. Each row reflects a different regression model, where the score for each psychopathology measure in the left column is the dependent variable, and inverse temperature parameters reflecting model-based ('MB Punish') and goal-perseveration ('GP Punish') punishment avoidance are the regressors. Each effect is presented in the following format: $\beta$ (standard error), p value. (**A**) Bivariate relationships without control covariates. (**B**) Regression coefficients when controlling for co-occurring levels of psychopathology as well as for general valence-independent levels of utilization of MB (inverse temperature and learning rate) and non-MB (AP, MF, and GP inverse temperatures) strategies. In all tables, p values are uncorrected for multiple comparisons.

particularly associated with difficulty in disengaging from worry (**Berenbaum, 2010**), since it lasts significantly longer in both clinical (**Dar and Iqbal, 2015**) and community samples (**Langlois et al., 2000**). Thus, we dissociated generalized from obsessive worry using the same approach taken in previous studies (**Doron et al., 2012**; **Stein et al., 2010**), namely, by including a measure of OCD symptoms as a control covariate. Controlling for OCD symptoms has the additional benefit of accounting for known relations between OCD and poor learning of task structure, reduced MB control, and perseverative tendencies (**Gillan et al., 2016**; **Seow et al., 2021**; **Sharp et al., 2021**). Secondly, another potentially confounding relationship exists between worry and somatic anxiety (**Sharp et al., 2015**), likely reflecting a general anxiety factor. Thus, we isolated worry by controlling for somatic anxiety, as commonly done in studies seeking to quantify distinct relationships of worry and somatic anxiety with cognitive performance (**Warren et al., 2021**) or associated neural mechanisms (**Silton et al., 2011**). Finally, we controlled for covariance between computational strategies that might reflect general task competencies. This included the utilization of MB (including learning rates and inverse temperatures) since observed anticorrelations in the empirical data (**Figure 4—figure supplement 4**) between GP and MB may derive from causal factors such as attention or IQ, as well as a general tendency to mitigate cognitive effort by using less costly strategies (AP, MF, and GP inverse temperatures; **Figure 4— figure supplement 4**).

This analysis showed a stronger relationship between worry and punishment perseveration ($\beta$ = 3.14 (1.38), $t$ = 2.27, p = 0.04, **Figure 5B**). No other significant relationship was observed between punishment perseveration or MB punishment avoidance and psychopathology (**Figure 5C**). Of note, we additionally found no association between the parameter governing how MB punishment was modulated by task block and levels of worry, both when including worry alone ($\beta$ = 2.5 (1.91), $t$ = 1.31, p = 0.19) and when controlling for the same covariates as detailed above ($\beta$ = 1.46 (1.65), $t$ = 0.88, p = 0.38). Ultimately, we validated the full model using a fivefold cross-validation procedure which showed

that regressing worry onto the aforementioned covariates (using a ridge regression implementation) explains significantly more variance in left out test-set data ($R^2$ = 0.24) relative to the models of the bivariate relationships between worry and GP Punishment ($R^2$ = 0.01) and MB Punishment ($R^2$ = 0.00).

It is important to note that all aforementioned p values testing our key hypotheses (*Figure 5B*) are corrected for multiple comparisons using a correction procedure designed for exploratory research (*Rubin, 2017*), which only controls for number of statistical tests within each hypothesis. Using a more conservative Bonferroni error correction for all four regression models, as typically employed in hypothesis-driven confirmatory work (*Frane, 2020*), resulted in a p value for the key effect of worry and punishment perseveration that no longer passed a conventional significance thresholds (p = 0.08). Thus, future work with a more targeted, hypothesis-driven approach needs to be conducted to ensure our tentative inferences regarding worry are valid and robust.

To illustrate the consequences of GP punishment avoidance on pursuit of reward in anxious participants, we simulated a GP + MB agent that adaptively engages reward-relevant information when instructed to, but perseverates in avoiding punishment during reward seeking. We show that such a strategy is as good as an MB agent in avoiding punishment, but comes with the cost of suboptimal reward seeking (*Figure 3B and C*). This trade-off mirrors the negative consequence of real-world threat avoidance in trait anxious individuals (*Aderka et al., 2012*). Moreover, this gives a potential normative explanation of punishment perseveration in anxious individuals; namely, if anxious individuals prioritize avoiding threat, they can do so just as well using punishment perseveration as using an MB strategy while expending fewer resources.

## Discussion

Using a novel multigoal pursuit task, we investigated computational strategies humans leverage to navigate environments necessitating punishment avoidance and reward seeking. Our findings indicate the use of a strategy that avoids goal switching wherein individuals learn a model of the task but use it in a goal-insensitive manner, failing to deactivate goals when they are irrelevant. This less flexible, but computationally less costly, strategy was leveraged more in order to avoid punishment as opposed to a pursuit of reward. Beyond trial-to-trial perseverance, inflexibility in punishment avoidance manifested in a lack of blockwise adjustment to the abundance of punishment goals. By contrast, we found that a flexible MB strategy was used more for reward seeking, and was flexibly modulated in an RR way in response to an abundance of reward seeking goals changing between task blocks. Finally, we demonstrate preliminary evidence that a greater GP reliance for punishment avoidance in those individuals with greater chronic worry.

The strategic deployment of GP primarily toward punishment avoidance indicates such behavior is not merely a reflection of a noisy or forgetful MB system. Our finding that humans use less flexible computational strategies to avoid punishment, than to seek reward, aligns with the idea of distinct neural mechanisms supporting avoidance and approach behavior (*McNaughton and Gray, 2000*; *Lang et al., 1998*). Moreover, comparative ethology and evolutionary psychology (*Pinker, 1997*) suggest there are good reasons why punishment avoidance might be less flexible than reward seeking. *Woody and Szechtman, 2011* opined that 'to reduce the potentially deadly occurrence of false negative errors (failure to prepare for upcoming danger), it is adaptive for the system to tolerate a high rate of false positive errors (false alarms).' Indeed, we demonstrate that in the presence of multiple shifting goals, perseverance in punishment avoidance results in false positives during reward seeking (*Figure 3B*), but avoids 'missing' punishment avoidance opportunities because of lapses in goal switching (*Figure 3C*). Future work could further test these ideas, as well as potential alternative explanations (*Dayan and Huys, 2009*).

GP may thus in fact constitute an RR strategy (*Lieder and Griffiths, 2019*) for approximating MB control. To illustrate this, consider that MB learning is computationally demanding in our task specifically because goals switch between trials. When the goals switch, an MB agent must retrieve and use predictions concerning a different feature. Additionally, the agent needs to continuously update its predictions concerning features even when they are not presently relevant for planning. GP avoids these computationally costly operations by pursuing goals persistently, thus avoiding switching and ensuring that features are equally relevant for planning and learning. In this way, GP saves substantial computational resources compared to MB yet is able to perform relatively well on the task, achieving better performance than MF. Additionally, if a participant selectively cares about

avoiding losses (for instance, due to loss aversion), GP can perform as well as MB. Thus, we propose the GP heuristic reflects a strategic choice, which can achieve good performance while avoiding the substantial resource requirements associated with MB control. In this sense it fulfils a similar role as other proposed approximations to MB evaluation including MF RL (*Sutton and Barto, 2018*), the successor representation (*Dayan, 1993*; *Momennejad et al., 2017*), mixing MB and MF evaluation (*Keramati et al., 2016*), habitual goal selection (*Cushman and Morris, 2015*), and other identified heuristics in action evaluation (*Daw and Dayan, 2014*).

Our exploratory findings that an inflexibility in punishment avoidance was more pronounced in individuals with chronic worry is suggestive of a computational deficit that may serve to unify several known effects relating trait worry to failure to terminate threat processing. For example, in paradigms that explicitly instruct participants to ignore threat-irrelevant information, such as the dot-probe (*Asmundson and Stein, 1994*) and modified emotional Stroop (*Dalgleish, 1995*; *van den Hout et al., 1995*) tasks, individuals with trait worry have difficulty inhibiting processing of threat (*Bar-Haim et al., 2007*). Moreover, increased GP punishment avoidance may be involved in the overactivity of threat-related circuitry in anxious individuals during tasks where threat is not present (*Grupe and Nitschke, 2013*; *Nitschke et al., 2009*). However, we note that there was a significant positive skew in the somatic arousal measure, which although likely due to random sampling error (given that other symptom measures were highly representative of population distributions), may nonetheless limit our ability to generalize findings from the present sample to the population.

Our findings go beyond previous findings that report, in single-goal reinforcement learning tasks, that anxiety is associated with altered MF but intact MB control (*Gillan et al., 2019*). Our findings suggest a conflict between punishment avoidance and reward seeking may be necessary to uncover how knowledge of task structure is used in anxiety. Indeed, prior approach–avoidance conflict paradigms have found that trait anxiety is positively associated with neural correlates of punishment avoidance (rejected gambles that could result in loss) (*Loh et al., 2017*) and avoidant behavior (*Bach, 2015*).

A limitation of our task is that differences in strategy utilization for reward seeking and punishment avoidance (see Methods) could in part reflect differences in sensitivity to reward versus punishment. However, reward and punishment sensitivity cannot account for the effects we observe across strategies, since on the one hand, punishment avoidance was greater for GP, whereas reward seeking was greater within an MB framework. Greater punishment sensitivity compared to reward sensitivity would predict the same direction of valence effect for both behavioral strategies. Moreover, knowledge of reward features had a greater net impact on choice across both goal-oriented strategies (sum of weights across both MB and GP strategies is greater for reward seeking). That said, we recognize that differences in outcome sensitivity, which are algorithmically equivalent to differences in the magnitude of external incentives, may cause a shift from one strategy to another (*Kool et al., 2017*). Thus, an open question relates to how reward and punishment sensitivity might impact flexibility in goal pursuit.

Future work can further address how humans manage multigoal learning in the context of larger decision trees with multiple stages of decisions. In such environments, it is thought people employ a successor feature learning strategy, whereby the long-run multistep choice features are stored and updated following feedback (*Tomov et al., 2021*). Such multistep tasks can be enhanced with shifting reward seeking and punishment avoidance goals to determine how altered strategies we identify with pathological worry might relate to trade-offs between MB and successor feature strategies for prediction. Another possibility is that punishment-related features capture involuntary attention in our task because they are tagged by a Pavlovian system, and this interacts with an MB system that learns task structure. Indeed, prior work (*Dayan and Berridge, 2014*) has discussed possibilities of MB Pavlovian hybrid strategies.

In relation to why GP punishment avoidance may specifically be associated with chronic worry, we suggest that failures to disengage punishment avoidance may serve to explain so-called 'termination' failures in chronic worry (*Berenbaum, 2010*). The causal role of GP in failures to terminate worry episodes could avail of the fact that such failures are dissociable from a tendency to suffer 'initiation' failures, which involve worrying too easily in response to many stimuli (*Berenbaum et al., 2018*). Although the perseveration of worry may appear relevant to obsessions in OC symptoms, punishment avoidance in OC has been empirically demonstrated to be specific to idiographic domains of potential threat (e.g., sanitation *Amir et al., 2009*), an issue the present study did not investigate. Additionally,

we did not find that GP was associated with somatic anxiety possibly due perhaps to random sampling error as we had an unusually low percentage meeting a threshold for mild symptomatology (4.7%; typical convenience samples are typically in the range of 12–18 *Sharp et al., 2015*; *Telch et al., 1989*). More importantly, somatic anxiety is thought to involve lower-order cognitive processes than those likely involved in multigoal pursuit (*Sharp et al., 2015*). Given that present results are preliminary in nature, future studies will need to test a prediction that chronic worry is associated with punishment perseveration in a larger sample. This should also include testing whether this association holds in a clinical population, as variation in symptoms in a clinical population may relate to punishment perseveration differently (*Imperiale et al., 2021*; *Groen et al., 2019*). Additionally, doing so may be enhanced by including parameters relating worry to punishment perseveration within the model-fitting procedure itself, and so better account for uncertainty in the estimation of such covariance (*Haines et al., 2020*).

In conclusion, we show humans are less flexible in avoiding punishment relative to pursuing reward, relying on a GP strategy that persists in punishment avoidance even when it is irrelevant to do so, and failing to deprioritize punishment avoidance goals when they are less abundant. Moreover, we show that GP is augmented in individuals with chronic worry, hinting at a candidate computational explanation for a persistent overprioritization of threat in anxiety.

## Materials and methods
### Sample and piloting
Prior to disseminating the task, we conducted a pilot study varying the number of features and actions participants could choose. We first found that less than half of individuals we recruited performed above chance levels when individuals had to learn a task with three actions and nine feature probabilities. We thus reduced the complexity of the task and found that including only two actions and four features allowed most participants to leverage an MB strategy.

Two hundred and forty-eight participants were recruited through Prolific services online Prolific recruiting service (https://www.prolific.co/) using the final task design from English-speaking countries to ensure participants understood task instruction. After expected data-scrubbing, our sample size had >99% power to detect valence differences in reinforcement learning parameters, conservatively assuming a medium effect size relative to large effects found previously (to account for differences between multigoal and single-goal settings; e.g., *Palminteri et al., 2017*; *Lefebvre et al., 2017*). Moreover, our sample had >80% power to detect small-medium effect sizes relating computational parameters and individual differences in anxiety ( *Sharp et al., 2021*). Participants gave written informed consent before taking part in the study, which was approved by the university's ethics review board (project ID number 16639/001). The final sample was 37% male, with a mean age of 33.92 years (standard deviation [SD] = 11.97). Rates of mild but clinically relevant levels of OCD (45%) and worry (40%) comported with prior studies (*Sharp et al., 2021*), indicating good representation of individual variation in at least some psychopathological symptoms.

### Data preprocessing
Eleven participants completed less than 90% of the trials and were removed. We next sought to define participants that did not understand task instructions. To do so, we computed the proportion of times participants switched from the last action taken as a function of the feature outcomes of that action and the current goal. We used these proportions to define four learning signals, two for each goal. Note, the term 'average' henceforth is shorthand for 'average switching behavior across all trials'. Facing reward goals, participants should (1) switch less than average if they observed a reward feature last trial and (2) switch more if they did not observe a reward feature last trial. Facing punishment goals, participants should (1) switch more than average if they observed a punishment feature last trial and (2) switch less than average if they did not observe a punishment feature last trial. We removed six additional participants because their switch behavior was the exact opposite as they should be for *each* of these four learning signals. When facing a reward goal, they switched more than average having observed a reward feature last trial and switched less than average having not observed a reward feature last trial. Moreover, when facing a punishment goal, they switched less having observed a punishment feature last trial and switched more than average having not observed

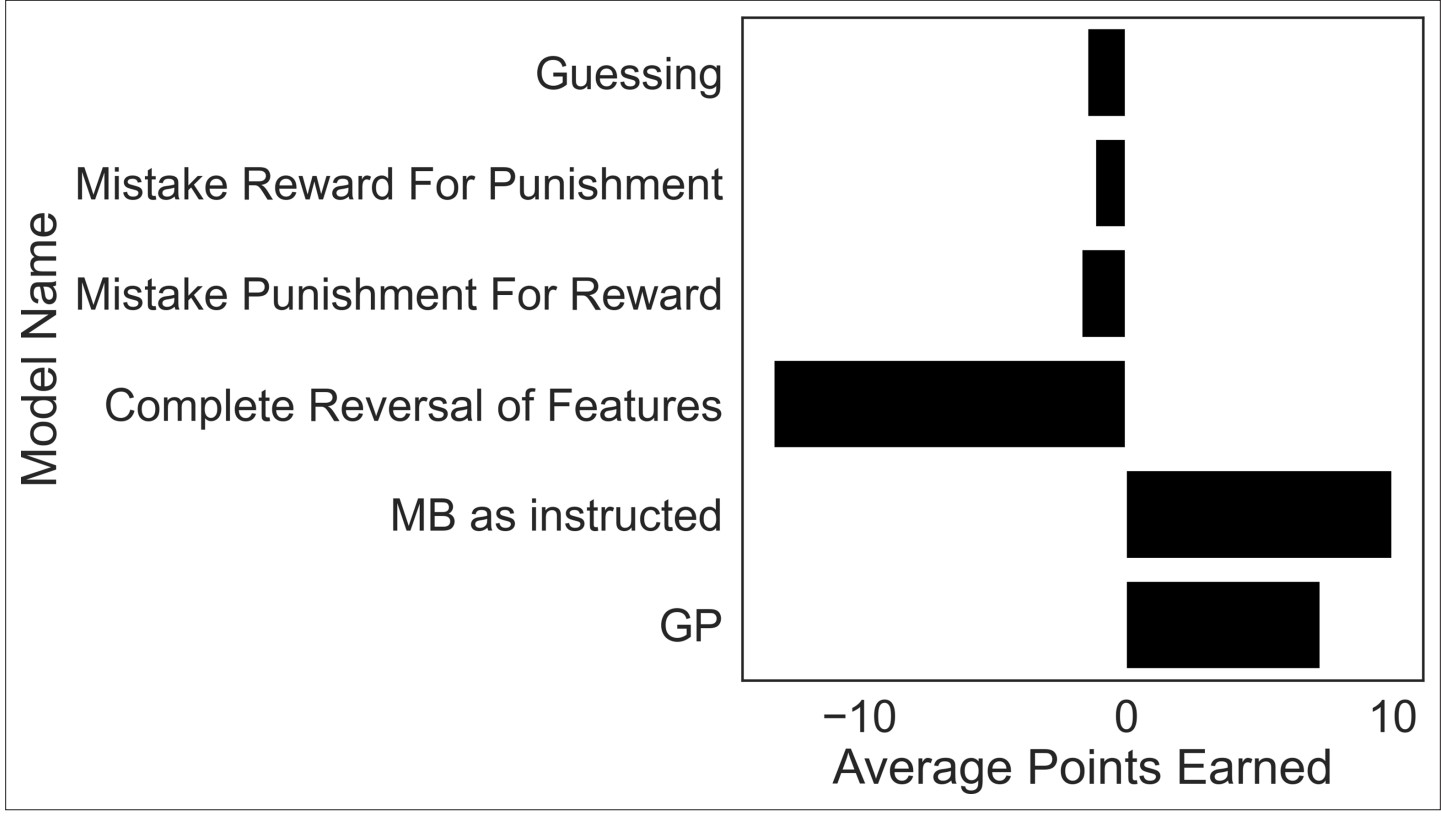

**Figure 6.** Excluded participants' strategies perform similar to or worse than purely guessing. To motivate our exclusion criteria, we simulated task performance by agents that falsify these criteria and calculated their average winnings over 5000 simulations each. The guessing agent chooses according to a coin flip. The models instantiating strategies used by excluded subjects comprise those that treat reward features as punishment features ('Mistake Reward for Punishment'), treat punishment features as if they were reward features ('Mistake Punishment for Reward') or incorrectly reverse the treatment of feature types ('Complete Reversal of Features'). Each model performed as poorly, or significantly worse, than a model that purely guesses, demonstrating a fundamental failure in these strategies for the present task. By contrast, a GP-only agent ('GP') that ignores goal-switching instructions does significantly better than a guessing model, and only a little worse than a pure model-based agent ('MB as instructed').

a punishment feature last trial. We additionally removed participants that: (1) treating a punishment feature as a reward feature (i.e., show the opposite of the two learning signals for punishment; 13 participants) and (2) treating reward feature as a punishment feature (show the opposite of the two learning signals for reward; 26 participants). Excluded subjects performed significantly worse in terms of choice accuracy. To derive accuracy, we computed the percentage of choices subjects made in line with an ideal observer that experienced the same outcome history (in terms of features) as each participant. On average, excluded subjects chose correctly 49.6% of time, whereas included subjects chose correctly 63.5% of time (difference between groups: $t(190) = 9.66$, $p < 0.00001$).

Indeed, these errors in following task structure are fundamental failures that result in average performance that is as poor or worse than an agent that purely guesses which action to take at each trial (*Figure 6*). Doing so resulted in our final sample of $n = 192$, with a low percentage of removed participants (22%) compared to similar online computational studies (*Tomov et al., 2021*). Importantly, removed participants were no different in terms of mean scores on key measures of psychopathology (greatest mean difference found in OCD; Welch's $t = -0.96$, $p = 0.33$).

Additionally, including such subjects would reduce our sensitivity to estimating differences in the utilization of GP and MB for goals of differing valence, as such subjects treated the task as if there was only a single goal, or that the goals were opposite of their instructed nature. Moreover, given their model of the task, such subjects could approach the task optimally using a MF strategy, and thus were not incentivized to use goal-directed strategies at all.

To determine whether our relatively strict subject exclusion policy might have affected the results, we conducted a sensitivity analysis on a larger sample ($n = 248$; 98% retention) including subjects that

mistreated the instructed value of certain features. To account for these subjects' behavior, we used normal priors to allow negative inverse temperature parameters. Fitting these revised models to our data, we again demonstrate that our winning model was the best-fitting model compared to all other models. Second, we show that the GP valence effect held and even came out stronger in this larger sample. Thus, the mean difference in GP utilization for punishment and reward goals was 0.24 in our original sample and 0.50 in the larger sample (p < 0.0001). Finally, we show the MB valence effect also held in this larger sample (original sample mean difference between MB reward and MB punishment = 2.10; larger sample mean difference = 1.27, both p values <0.0001).

## Symptom questionnaires

Before participants completed the online task, they filled out three questionnaires covering transdiagnostic types of psychopathology. Chronic worry was measured via the 16-item Penn State Worry Questionnaire (*Meyer et al., 1990*). Anxious arousal was measured with the 10-question mini version of the Mood and Anxiety Symptom Questionnaire – Anxious Arousal subscale (*Casillas and Clark, 2000*). Obsessive compulsiveness was measured with the Obsessive Compulsive Inventory – Revised (*Foa et al., 2002*).

## Multigoal pursuit task

To examine how people learn and utilize knowledge about the outcomes of their actions in order to seek reward and avoid punishment, we designed a novel multigoal pursuit task. The task consisted of 160 trials. On each trial, participants had 4 s to make a choice between two choice options (represented as fractal images; (*Figure 1B*)). Choosing a fractal could then lead to two independent outcome features, a gold feature and a black feature. Any given choice could thus lead to both features, neither feature, or one of the two features. The chances that a choice led to a certain feature varied according to slowly changing probabilities (*Figure 1A*). These probabilities were partially independent of one another (i.e., the rank correlation between any pairs of correlation did not exceed 0.66; full span: [0.02,−0.17,0.24,0.28,−0.66,0.43]). The same sequence of feature probabilities (probability of encountering a given feature conditioned on a given choice) was used for all participants. This sequence was generated by starting each sequence at a random value between 0.25 and 0.75, and adding independent noise (normally distributed mean = 0.0, SD = 0.04) at each trial to each sequence, yet bounding the probabilities to be between 0.2 and 0.8. To incentivize choice based on feature probabilities, we ensured that in resultant sequences the probability of reaching a given feature differed between the two choices by at least 0.27 for each feature, on average across the 160 trials.

To manipulate participant's goals, throughout the task, one of the two outcome features (which we refer to as the reward feature) either provided 1 or 0 point, and the other outcome feature (which we refer to as the punishment feature) would either provide 0 points or take away one point. The number of points that a given outcome feature provided on a given trial was determined by trial-specific instructed goals (on the screen on which choice options were presented, *Figure 1B*). A punishment avoidance goal meant the punishment feature took on the value of −1 and the reward feature took on the value of 0, whereas the reward seeking goal meant the punishment feature took on a value of 0 and the reward feature took on the value of +1. This information was presented in a color (gold or silver) matching the corresponding outcome feature's color. Importantly, the color the feature took on and the probability trajectory for either feature were counterbalanced across participants. Participants were instructed in advance that one feature in the task might tend to provide points and the other feature might tend to take away points, but they were not told which features these would be.

To manipulate goal abundance, the frequency of which feature was set to the nonzero outcome varied in the first half versus the second half of the experiment (*Figure 1C*). In one half of the experiment (punishment context), the punishment feature took away a point (and the reward feature did not provide a point) on 65% of trials, and in the other half (reward context) the reward feature provided a point on 65% of trials. Which context occurred in the first versus second half of the experiment was counterbalanced across participants.

After the participant observed which outcome features of a choice they received (2 s), they observed the number of points they received (1 s), including the individual reward value of each feature received, in white, as well as a sentence summarizing their total earnings from that trial (e.g., 'You lost 1'). Following this, in order to eliminate the need for exploratory choices, the participant was

shown the features they would have received, had they chosen the alternative choice option (2 s). There was then a 1-s intertrial interval.

## Clinical analyses

Although the computational parameters were nonindependently estimated by our hierarchical model-fitting procedure, it is vital to note this does not compromise the validity of the least-squares solution to the regressions we ran. Indeed, Friedman, *Hastie et al., 2009* show that, 'Even if the independent variables were not drawn randomly, the criterion is still valid if the dependent variables are conditionally independent given the [independent variable] inputs' (p. 44). However, we note that it is in practice difficult to determine whether such conditional independence is met. In each regression, we excluded the learning rate for counterfactual feedback, as well as learning rate for value in the MF system, due to high collinearity with other parameters (see Methods). We verified low multicollinearity among included parameters (variance inflation factor <5 for independent variables *Akinwande et al., 2015*). We report all bivariate correlations between fitted parameters in *Figure 4—figure supplement 4*.

## Algorithms defining MB, MF, and GP strategies

We first describe how each learning system in the winning model derived action values. Then, we describe how action values were integrated into a single decision process. Together, these comprise the best-fitting model that we report on in Results.

### MB system

An MB agent learns each of the four semi-independent transition probabilities of reward and punishment features given either of the two actions one can choose. Each trial, this agent updates their prior estimate of observing a feature given an action (either 'press $g$' or 'press $j$') incrementally via a feature prediction error and a learning rate, $\alpha_{chosen}$. Here, an agent pressed '$g$', observed a punishment feature, and updated the probability of observing a punishment feature conditional on choosing to press '$g$':

$$
\begin{aligned}
P\left(f = punish | press = g\right)_{t+1} = \\
P\left(f = punish | press = g\right)_t + \alpha_{chosen}(1 - P(f = punish | press = g)_t)
\end{aligned}
\tag{1}
$$

Here, the '$t$' subscript refers to the trial, and '1' in the parentheses means that the participant observed a punishment feature. If the feature was not present, the absence would be coded as a '0'. This same coding (one for feature observation, 0 if absent) was also used to encode the presence or absence of a reward feature.

The model learns similarly from counterfactual feedback, albeit at a different rate. Thus, at each trial, MB agents update feature probabilities for the action they did not choose via the same equation as above but with learning rate $\alpha_{unchosen}$. If the agent pressed '$g$' the counterfactual update would be:

$$
\begin{aligned}
P\left(f = punish | press = j\right)_{t+1} = \\
P\left(f = punish | press = j\right)_t + \alpha_{unchosen}\left(1 - P\left(f = punish | press = j\right)_t\right)
\end{aligned}
\tag{2}
$$

Each of the four probabilities an MB agent learns is stored in a matrix, where rows are defined by actions and columns by feature type (i.e., reward or punishment). These stored feature probabilities are multiplied by 'utilization weights' ($\beta_{MBpunish}$ and $\beta_{MBreward}$) that reflect the degree to which an agent utilizes an MB strategy to pursue reward or avoid punishment. No additional parameter controls utilization of an MB strategy (e.g., there is no additional overall $\beta_{MB}$).

Each trial, the agent computes the expected value of each outcome by multiplying stored feature probabilities given each action with the values of the features that are defined by the trial-specific goal. Here, the agent is facing an avoid punishment trial, for which the presence of a punishment feature results in taking away one point (i.e., a value of −1; below we abbreviate press as '$p$', reward as 'rew', and punishment as 'pun'):

$$
\begin{bmatrix} Q_{MB\,t+1}\,(p=g) \\ Q_{MB\,t+1}\,(p=j) \end{bmatrix} = \begin{bmatrix} \beta_{MBreward}P\,(f=rew|p=g)_t & \beta_{MBpunish}P\,(f=pun|p=g)_t \\ \beta_{MBreward}P\,(f=rew|p=j)_t & \beta_{MBpunish}P\,(f=pun|p=j)_t \end{bmatrix} \begin{bmatrix} 0 \\ -1 \end{bmatrix} \tag{3a}
$$

Via this computation an MB agent disregards the irrelevant goal (here reward seeking). If the agent were facing a reward goal, *Equation 3a and 3b* would be:

$$
\begin{bmatrix} Q_{MB\,t+1}\,(p=g) \\ Q_{MB\,t+1}\,(p=j) \end{bmatrix} = \begin{bmatrix} \beta_{MBreward}P\,(f=rew|p=g)_t & \beta_{MBpunish}P\,(f=pun|p=g)_t \\ \beta_{MBreward}P\,(f=rew|p=j)_t & \beta_{MBpunish}P\,(f=pun|p=j)_t \end{bmatrix} \begin{bmatrix} 1 \\ 0 \end{bmatrix} \tag{3b}
$$

## Resource reallocation
Within the MB system, utilization weights changed across block according to a change parameter. Below is an example of how this reallocation occurred for $\beta_{MBreward}$ :

$$
\beta_{MBreward,block} = \begin{cases} \beta_{MBreward} + \beta_{changeReward} & \text{if } reward\_rich\, block \\ \beta_{MBreward} - \beta_{changeReward} & \text{if } punishment\_rich\, block \end{cases} \tag{4}
$$

Note that negative $\beta_{change}$ values were allowed, and thus the model did not assume a priori that, for instance, individuals would have increased MB control for reward in the rewarding block (it could be the opposite). Thus, if the data are nevertheless consistent with a positive $\beta_{change}$ , this is an indication that, although participants were not explicitly told which block they were in, they tended to prioritize the more abundant goal in each block (see 'Alternative models' for an attempt to model how participants inferred goal frequency).

## MF system
A MF agent learns the value of either action directly based on the reward and punishment received. In our task, outcomes took on values of {−1,0,1}. Action values were updated incrementally via a value prediction error and learning rate for value, $\eta$. Below is an example updating the action value for press = j (which we omit from the right side of the equation for concision):

$$
Q_{MF\,t+1}\,(press=j) = Q_{MF\,t} + \eta\,\left(OutcomeValue_t - Q_{MF\,t}\right) \tag{5}
$$

## Goal perseveration
A GP agent uses the same matrix of estimated feature probabilities as the MB system, but multiplies them by a static vector, $\begin{bmatrix} 1 \\ -1 \end{bmatrix}$, which means the system always engages both goals regardless of instructions. This is the only way in which the GP agent differs from the MB agent. Having its own utilization weights ($\beta_{GPreward}$ and $\beta_{GPpunish}$) allows the system to vary across individuals in the degree to which the 'avoid punishment' and 'seek reward' goals are each pursued when they are irrelevant:

$$
\begin{bmatrix} Q_{GP_{t+1}(press=g)} \\ Q_{GP_{t+1}(press=j)} \end{bmatrix} = \begin{bmatrix} \beta_{GPreward}P\,(f=reward|a=press\,g)_t & \beta_{GPpunish}P\,(f=punish|a=press\,g)_t \\ \beta_{GPreward}P\,(f=reward|a=press\,j)_t & \beta_{GPpunish}P\,(f=punish|a=press\,j)_t \end{bmatrix} \tag{6}
$$

Note, we also fit a model where the GP strategy learns its own matrix of estimated feature probabilities separate from that learned by the MB strategy (i.e., with a different learning rate), but this did not fit participants' choices as well (*Figure 4—figure supplement 3*.).

## Action perseveration
Actions taken on the last trial were represented by a one-hot vector (e.g., $\begin{bmatrix} 1 \\ 0 \end{bmatrix}$), which we store in a variable, $Q_{LastTrial}$ which was multiplied by its own utilization parameter, $\beta_{AP}$ .

## Stochastic decision process

All action values derived by each system were integrated via their utilization weights. Below we show the integrated action value for press = $j$ (which we omit from the right side of the equation for concision):

$$Q_{Integrated}\ (press = j) = Q_{MB} + Q_{GP}\ + \beta_{MF}Q_{MF}\ + \beta_{AP}Q_{LastTrial} \qquad (7)$$

Note there are no utilization weights in the above equation for MB and GP $Q$ values because they were already integrated into these $Q$ values in *Equations 3a, 3b and 6*. The integrated action value was then inputted into a softmax function to generate the policy, which can be described by the probability of pressing '$j$' (since the probability of pressing '$g$' is one minus that):

$$P\left(press = j\right) = \frac{e^{Q_{Integrated}\left(press=j\right)}}{e^{Q_{Integrated}\left(press=j\right)} + e^{Q_{Integrated}\left(press=g\right)}} \qquad (8)$$

## Alternative models tested

We tested several alternative models that did not explain the data as well as the winning model described above (full details in *Figure 4—figure supplement 3*). First, we tested models that included only one of the strategies described above (i.e., only MB, only GP, and only MF). We then tested models in a stepwise fashion detailed in *Figure 2*, which demonstrated that adding each strategy contributed to explaining unique variance in the data.

We additionally tested a model where differences in reward seeking and punishment avoidance were captured by the learning process as opposed to the utilization of the learned knowledge. To do so, we endowed the model with different MB and GP learning rates for punishment and reward features ($\alpha_{MBreward}$, $\alpha_{MBpunish}$, $\alpha_{GPreward}$, $\alpha_{GPpunish}$) in *Equation 6*, and a single utilization weight ($\beta_{MB}$, $\beta_{GP}$).

With regard to the MB strategy, we additionally tested a model where learning from counterfactual outcomes was implemented with the same learning rate as learning from the outcomes of one's actions.

With regard to resource reallocation, we additionally tested models where it occurred in just the GP utilization weights, or in both GP and MB utilization (in the same fashion described in *Equation 4*). After finding that data were best explained by the model where resource reallocation only occurred in the MB system, we tested if resource reallocation changed from trial to trial as function of recently experienced goals. That is, we examined whether individuals recruit more resources toward the goal one has most recently experienced, which could differ within a given task block.

With regard to the MF strategy, we tested a model where goals were encoded as part of its state representation (G-MF). Specifically, action values were learned separately for trials with punishment avoidance goals ($Q_{G-MFpunish, press\ j}$ and $Q_{G-MFpunish, press\ g}$) and reward seeking goals ($Q_{G-MFreward, press\ j}$ and $Q_{G-MFreward, press\ g}$). In this version of an MF strategy, experienced outcomes only influence decision-making under the goal in which they were experienced. The main way it differs from an MB strategy is that learning relevant to a particular goal occurs *only when that goal is active*. Thus, $Q$ values cannot track feature probabilities changing during nonmatched goal trials (e.g., how reward feature probabilities might shift during punishment avoidance trials). This may be one reason why it was inferior to the best-fitting model. Similar to the best-fitting model, this model included separate utilization weights ($\beta_{G-MFreward}$ and $\beta_{G-MFpunish}$ and a single learning rate. GP and perseveration strategies were included as in the best-fitting model, and resource reallocation was applied to the G-MF strategy in the same way as described in *Equation 4*.

## Model fitting

Models were fit with a hierarchical variant of expectation–maximization algorithm known as iterative importance sampling (*Bishop, 2006*), which has been shown to provide high parameter and model recoverability (*Michely et al., 2020b*; *Michely et al., 2020a*). The priors for this model-fitting procedure largely do not affect the results, because the procedure iteratively updates priors via likelihood-weighted resampling in order to converge on the distributions of parameters that maximize the integrated BIC, an approximation of model evidence. Note, all parameters had weakly informed priors (see *Figure 2—figure supplement 2*). Specifically, the fitting procedure works by (1) drawing 100,000

samples from all group-level parameter distributions for each participant, (2) deriving participant-level likelihoods for each sampled parameter, (3) resampling parameters after weighting each sample by its likelihood, and (4) fitting new hyperparameters to the overall distribution of resampled parameter values. This process continues iteratively until the integrated BIC of the new parameter settings does not exceed that of the last iteration's parameter settings.

### Model and parameter recoverability

To verify that the experiment was appropriately designed to dissociate between the tested models and their parameter values, we simulated experimental data from the best-fitting and reduced models and successfully recovered key inverse temperature parameters (all above 0.58 correlation, average correlation = 0.79). The model that generated the data was recovered 10/10 times compared to the next best-fitting model (see *Figure 4—figure supplement 2*).

### Simulating mechanism-agnostic stay-switch behavior

In order to examine model predictions (*Figure 2*), we used each given model to simulate experimental data from 400 participants, each time generating a new dataset by setting model-relevant beta parameters to 5, learning rate parameters to 0.2, and other parameters to 0. We then computed the proportion of trials in which the model chose a different action compared to the previous trial. This 'switch probability' was computed for each combination of the previous and current trials' goals, and the features observed on the previous trial. We ensured there were no significant differences in the direction and significance of key effects across task versions by separately fitting our Bayesian logistic regression noted above to the subset of subjects that performed each task version. Doing so showed that all effects held and to a remarkably similar degree in both task versions (see full results in *Supplementary file 1*).

### Simulating the optimality of each computational strategy

We simulated artificial agents playing the exact task 2000 times and plotted the mean reward earned. Each artificial agent was also endowed with a learning rate for feature probabilities, which sampled from a grid of values over the 0–1 range at 0.02 increments. For each type of agent, we set the utilization weights of the relevant strategy to five and other utilization weights (for nonused strategies) to 0.

### Testing differences between reward seeking and punishment avoidance parameters

As a consequence of the iterative nature of the model-fitting procedure, parameters for individual participants are not independently estimated, precluding the use of Bayesian or frequentist parametric tests. We thus used nonparametric tests to compute unbiased p values. Due to a heavy positive skew in the distributions of utilization weight parameters at the group level (*Figure 4B and C*), we compared between them in terms of their median levels. We note that the skew in inverse temperature parameters is to be expected given their Gamma prior distributions are inherently skewed (*Gillan et al., 2016*; *Sharp et al., 2021*). Thus, we generated null distributions of median differences in utilization weights for both MB and GP strategies. To do so, we ran our hierarchical model-fitting procedure 300 times on 300 simulated datasets that assumed reward and punishment utilization weights were sampled from the same distribution (null hypothesis). The utilization weights that best fitted the simulated data were used to generate the null distribution of median differences. We then generated p values for median differences in the empirical data by seeing how extreme the empirical value was with respect to the generated null distribution. Each simulated dataset comprised the same number of participants as in the original sample ($n$ = 192) and sampled both parameters with replacement from a joint distribution representing the null hypothesis that the two parameters are equal. The null distribution was derived through running our model-fitting procedure on the empirical data for one iteration to derive true posteriors at the participant level, and combining the participant-level median parameter estimates for both parameters of interest (e.g., utilization parameters for MB reward and MB punishment) to form a joint distribution. All other parameters were drawn from the group-level distributions derived by fitting the winning model to participants' data.

### Testing significance of resource reallocation parameters

We tested the difference of resource reallocation parameters ($\beta_{changeReward}$ and $\beta_{changePunishment}$ in *Equation 4*) from zero using a permutation test, wherein we generated a null distribution by shuffling

the labels denoting task block within each participant and recording the mean for each change parameter. We then generated p-values for the empirical means of change parameters by computing the proportion of the null distribution exceeding the empirical value.

## Behavioral signatures Bayesian logistic regression

The regression sought to explain whether participants stayed (coded as 1) or switched (coded as 0) on trial '$t$', which we refer to as '$choice_t$' in **Equation 9**, for the subset of trials where the current goal differed from the goal encountered on the previous trial. Since current and previous goals are perfectly anticorrelated in such trials, the main effect of goal was simply encoded as:

$$
\text{goal}_{t-1} = \begin{cases} 1 & \text{if last goal = reward seeking} \\ -1 & \text{if last goal = punishment avoidance} \end{cases}
$$

GP effects were modeled by variables that encoded whether features were observed for both chosen and unchosen actions last trial with the following encoding scheme (here for reward):

$$
\text{rwd}_{t-1} = \begin{cases} 1 & \text{if reward feature observed only for chosen action} \\ 0 & \text{if outcomes were the same for chosen and unchosen actions} \\ -1 & \text{if reward feature observed only for unchosen action} \end{cases}
$$

MB effects were modeled by the interaction of the GP terms and the current goal as follows (here again for the MB reward effect):

$$
\text{rwd}_{t-1} \, X \, goal_t = \begin{cases} 1 & \text{if rwd}_t = 1 \text{ and goal}_{t-1} = -1 \\ 0 & \text{if goal}_{t-1} = 1 \\ -1 & \text{if rwd}_t = -1 \text{ and goal}_{t-1} = -1 \end{cases}
$$

Last, we modeled MF effects as the interaction between reward and punishment features observed for *chosen* actions and the last goal faced (here, for MF reward effects):

$$
\text{rwd}_{chosen,t-1} \, X \, goal_{t-1} = \begin{cases} 1 & \text{if rwd}_{t,chosen} = 1 \text{ and goal}_{t-1} = 1 \\ 0 & \text{if goal}_{t-1} = -1 \\ -1 & \text{if rwd}_{t,chosen} = -1 \text{ and goal}_{t-1} = 1 \end{cases}
$$

The dissociation between this MF signature and the MB signature described above relies on the insensitivity of the MF strategy to counterfactual outcomes, which possess no present value.

We included all independent variables in a Bayesian mixed-effects logistic regression as follows:

$$
p(choice_t = 1) = \text{logistic}(\beta_1 intercept + \beta_2 goal_{t-1} + \overbrace{\beta_3 rwd_{t-1} + \beta_4 pun_{t-1}}^{goal\ perseveration} + \\ \overbrace{\beta_5 rwd_{t-1} X\, goal_t + \beta_6 pun_{t-1} X\, goal_t}^{model-based} + \\ \underbrace{\beta_7 rwd_{chosen,t-1} X\, goal_{t-1} + \beta_8 pun_{chosen,t-1} \, X\, goal_t}_{Model-free})
$$

(9)

Posterior probability distributions of each effect were estimated using a sampling procedure in BAyesian Model-Building Interface (Bambi) in Python (**Capretto et al., 2020**), which is a high-level interface using the PyMC3 Bayesian modeling software. The default sampler in Bambi an adaptive dynamic Hamiltonian Monte Carlo algorithm, which is an instance of a Markov chain Monte Carlo sampler. In all models, all estimated effects had good indicators of reliable sampling from the posterior, including r-hat below 1.1 and effective sample size above 1000 for all parameters. Note, **Equation 9** is written at the participant level. Each effect was drawn from a normal group distribution the mean and variance of which were drawn from prior distributions, estimated by Bambi's default algorithm, which is informed by implied partial correlations between the dependent and independent variables, and has been demonstrated to produce weakly informative but reasonable priors (**Capretto et al.,**

*2020*). For hypothesis testing, we compared the 95% most credible parameter values (i.e., the 95% highest density intervals) to a null value of 0.

## Acknowledgements

PBS is supported by a Fulbright postdoctoral fellowship. EE is supported by NIH grants R01MH124092 and R01MH125564, ISF grant 1094/20 and US Israel BSF grant 2019801. RJD holds a Wellcome Trust Investigator award (098362/Z/12/Z). The Max Planck UCL Centre for Computational Psychiatry and Ageing Research is a joint initiative supported by the Max Planck Society and University College London.

## Additional information

### Funding

| Funder | Grant reference number | Author |
| --- | --- | --- |
| Fulbright Association | PS00318453 | Paul B Sharp |
| Israel Science Foundation | 1094/20 | Eran Eldar |
| Wellcome Trust | 098362/Z/12/Z | Raymond J Dolan |
| National Institutes of Health | R01MH124092 | Eran Eldar |
| National Institutes of Health | R01MH125564 | Eran Eldar |
| Israel Binational Science Foundation | 2019801 | Eran Eldar |

The funders had no role in study design, data collection, and interpretation, or the decision to submit the work for publication.

### Author contributions

Paul B Sharp, Conceptualization, Data curation, Formal analysis, Investigation, Methodology, Visualization, Writing – original draft, Writing – review and editing; Evan M Russek, Conceptualization, Formal analysis, Methodology, Writing – original draft, Writing – review and editing; Quentin JM Huys, Supervision, Writing – review and editing; Raymond J Dolan, Funding acquisition, Supervision, Writing – review and editing; Eran Eldar, Conceptualization, Formal analysis, Supervision, Writing – original draft, Writing – review and editing

### Author ORCIDs

Paul B Sharp ⬤ http://orcid.org/0000-0003-4949-1501
Raymond J Dolan ⬤ http://orcid.org/0000-0001-9356-761X

### Ethics

Human subjects: Participants gave written informed consent before taking part in the study, which was approved by the university's ethics review board (project ID number 16639/001).

### Decision letter and Author response

Decision letter https://doi.org/10.7554/eLife.74402.sa1
Author response https://doi.org/10.7554/eLife.74402.sa2

## Additional files

### Supplementary files

• Supplementary file 1. Results of fitting two separate hierarchical Bayesian logistic regression models to empirical data from each version of the task.

• Transparent reporting form

## Data availability

All data are available in the main text or the supplementary materials. All code and analyses can be found at: https://github.com/psharp1289/multigoal_RL, copy archived at swh:1:rev:1cf24428da17e8bcb2fab6d0ff9a7a59ee1586f7.

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
