## [Decision Letter]

**Decision letter after peer review:**

Thank you for submitting your article "Humans perseverate on punishment avoidance goals in multigoal reinforcement learning" for consideration by *eLife*. Your article has been reviewed by 3 peer reviewers, including Claire M Gillan as Reviewing Editor and Reviewer #3, and the evaluation has been overseen by Christian Büchel as the Senior Editor.

The reviewers were overall favourably disposed to this manuscript – it was felt that the paper extends previous work on how humans use a mix of learning strategies to make decisions in cognitively demanding environments to show that humans flexibly adjust goal-directed learning based on statistics of the environment but inflexibly engage in punishment avoidance regardless of current goals. The task design and model are novel and interesting. However, all 3 reviewers had substantial concerns (i) aspects of the behavioural analysis (modelling), (ii) the interpretations of behavioural effects, and perhaps most substantially, (iii) the extension to psychopathology. I have outlined these essential revisions below by way of summary. Individual reviewers made further suggestions that can be considered optional, though we encourage that they also be addressed as they will strengthen the paper.

Essential revisions:

1) All three reviewers felt that the clinical analyses, albeit interesting, were not approached with the same rigour and clarity as the analyses of behaviour preceding them (which were excellent). A revision must address this comprehensively, including (i) how non-independence of parameter estimates influence clinical analyses, (ii) the inclusion of clinical covariates without first reporting bivariate effects, (iii) correction for multiple comparisons should be more extensive, (iv) visualisation of all clinical effects not just select ones, (v) more complete reporting with respect to all clinical measures gathered (including analysis of somatic anxiety). Related to this, the authors may wish to consider in the discussion that there are potential difference between non-clinical and clinical cohorts in terms of correlation of clinical measures, though the data are still a bit mixed:

Imperiale, M. N., Lieb, R., Calkins, M. E., and Meinlschmidt, G. (2021). Multimorbidity networks of mental disorder symptom domains across psychopathology severity levels in community youth. Journal of psychiatric research, 141, 267-275.

Groen, R. N., Wichers, M., Wigman, J. T., and Hartman, C. A. (2019). Specificity of psychopathology across levels of severity: a transdiagnostic network analysis. Scientific reports, 9(1), 1-10.

2) With respect to the modelling of behaviour, (i) more information about parameter recovery is requested, and (ii) concerns were raised about parameter estimation more generally given the skew evident in the inverse temperatures, coupled with the multicollinearity. More information should be provided here, including proportion of recovered parameters that can capture the main parameter-based results of the paper.

3) The interpretation of the key behavioural effects: could the authors defend the interpretation that GP is not an actual strategy, but rather a noisy or distracted attempt at MB learning. It was felt that it is rather a big claim to say that the GP approach is a reasonable third strategy on top of MF/MB learning and if this interpretation is to be maintained, the authors need to back this up by showing it is something more than just an imprecise MB approach. This ties in with additional concerns about the exclusion of participants that did not obey certain rules of the task. It was not clear how their exclusion was justified given a valid interpretation of some of these key effects may be that (i) people find it hard to keep instructions in mind in complex tasks, (ii) people may be utilising strategies that you have not defined and are not well understood but are nonetheless real – e.g. belief that reward probabilities / 'luck' switches from trial to trial. Other concerns regarding exclusion were that they appear to be asymmetric with respect to reward/punishment conditions, suggesting these data are meaningful.

*Reviewer #1 (Recommendations for the authors):*

1. Psychopathology analyses.

a. If the authors wish to make a connection to psychopathology, reporting the relationship between worry alone – rather than controlling for other overlapping symptom measures and model parameters – would be more appropriate. The recommendations for testing multiple, partially overlapping psychopathology measures in this paper may be helpful: DOI: 10.1177/21677026211017834

b. An alternate approach would be to focus this paper on the main findings about learning strategies and to save relationships to psychopathology for a future paper with a more appropriate sample.

2. Parameter-based analyses.

a. Providing more information on parameter recovery is needed. In particular, showing the proportion of recovered parameters that can capture the main parameter-based results of the paper (Figure 2C/D) would show that these findings reflect true underlying parameter differences rather than artifacts of model estimation.

b. If the authors retain the psychopathology analyses, they should be conducted in a way that does not assume independence of parameter estimates.

c. Alternatively, the analyses using relative model fits and trialwise regressions provide most of the information needed for the conclusions of the paper. The parameter-based analyses could be omitted with the focus instead on these other kinds of analyses.

*Reviewer #2 (Recommendations for the authors):*

The authors used the term "predicted" quite a bit to describe associations. I don't think this is justified (they haven't really done any predictive analyses).

If I understand correctly, the same 4 random walks were used for all participants (randomised between the 4 associations). Of the two shown, one looks much more stable than the other. It would be useful to see all 4 walks to see how comparable they are (if I am correct that the same 4 are used for all participants). If the walks are very different, should their allocation to the associations be controlled for in the analysis?

It would be useful to report the relationship between worry and the block effect (i.e. you suggest high worry is associated with higher GP/lower MB for losses-do worried people adapt to changes in the base rates of the outcomes?).

*Reviewer #3 (Recommendations for the authors):*

Well done on an interesting read and a contribution that will be informative for a lot of researchers. I have some suggestions to improve the paper.

All analyses with the 3 clinical factors should be presented in full; including supplementary figures if possible. Simple associations should be carried out before adding covariates to assist the reader in interpreting these findings and in generating hypotheses based on them. OCD is said to be not related to parameters at p=.08, while worry is at p=0.04 (uncorrected i guess more like p=0.02 for the latter), these are not likely to be different from one-another. And they may depend on the inclusion of these variables in the same model. Reader needs more transparency around these effects and any claims of specificity need more support. The data presented actually suggests the opposite.

Relatedly, the result in relation to worry, the effect is marginal at p=.04. While 2 multiple comparisons are controlled for, this is a fairly liberal decision given several tests were conducted and reported (i.e. GP MB and MF for punishment/reward = 6 at least; plus the 3 clinical scales = 18 etc). I'd encourage the authors to report all of the associations in a table, correct for multiple comparisons. This will serve the same purpose of suggesting the most interesting avenue for future research but also give the reader a fuller view on specificity of this to worry. This exploratory framing for the clinical effects does not detract from the main contribution of the paper or the potential for this to be especially interesting for 'worry' – it would just make them clearer and let the reader decide that for themselves a bit more.

There needs to be a bit more done with respect to relating the clinical variables to the model parameters. I would have thought this would be best placed within the hierarchical model itself. Alternatively, I wonder if these is a point-estimate that could be generated that is more flexible and less dependent on the overall group effects and other parameter values.

The authors describe issues with collinearity of the parameter values. Can a correlation matrix in the supplement be included that reports these (I think currently you can sort of see it based on simulated vs real data, but this is not the same as correlating real vs real across params).

I strongly encourage all subjects are retained (though i feel less strongly about excluding those not completing enough trials, 90% even seems a bit harsh/wasteful of data). If not, then a clear justification for why the strategy or approach of these subjects is not an accurate reflection of potentially the decision making preferences of 22% of the population. More standard indicators of inattentive responding focus on RTs, overly rigid responding that renders modelling choice impossible. Not clear why these were not used here as they seem better justified indicators of inattentive subjects. At the risk of belabouring the point(!), defining these subjects as 'not understanding instructions' could be applied to many of the key findings of this paper (i.e. avoidance perseveration suggests they don't pay attention to the current goals etc). So I think this practice is not ideal.

---

## [Author Response]

Essential revisions:1) All three reviewers felt that the clinical analyses, albeit interesting, were not approached with the same rigour and clarity as the analyses of behaviour preceding them (which were excellent). A revision must address this comprehensively, including (i) how non-independence of parameter estimates influence clinical analyses, (ii) the inclusion of clinical covariates without first reporting bivariate effects, (iii) correction for multiple comparisons should be more extensive, (iv) visualisation of all clinical effects not just select ones, (v) more complete reporting with respect to all clinical measures gathered (including analysis of somatic anxiety). Related to this, the authors may wish to consider in the discussion that there are potential difference between non-clinical and clinical cohorts in terms of correlation of clinical measures, though the data are still a bit mixed:Imperiale, M. N., Lieb, R., Calkins, M. E., and Meinlschmidt, G. (2021). Multimorbidity networks of mental disorder symptom domains across psychopathology severity levels in community youth. Journal of psychiatric research, 141, 267-275.Groen, R. N., Wichers, M., Wigman, J. T., and Hartman, C. A. (2019). Specificity of psychopathology across levels of severity: a transdiagnostic network analysis. Scientific reports, 9(1), 1-10.

We thank the reviewers and editor for thoughtful comments, including suggestions for improving the clarity and comprehensiveness of our clinical analyses. Our approach to addressing this set of concerns is twofold: Firstly, we followed Reviewer 3’s suggestion (see Reviewer comment 3.7) to re-frame the clinical analyses as exploratory, requiring further testing before definitive conclusions can be drawn. Secondly, in accordance with the reviewers’ suggestions, we substantially expanded the breadth and clarity of the reported analyses. We next detail these two sets of modifications.

A. Re-framing clinical analyses and exploratory and preliminary

To clarify that present clinical analyses are exploratory, and further investigation is required to test the validity of the findings, we implemented the following changes to the text:

I. A revision of the Abstract to make clear that inferences about worry and punishment perseveration are preliminary:

**“**Importantly, we show preliminary evidence that individuals with chronic worry may have difficulty disengaging from punishment avoidance when instructed to seek reward. Taken together, the findings demonstrate that people avoid punishment less flexibly than they pursue reward. Future studies should test in larger samples whether a difficulty to disengage from punishment avoidance contributes to chronic worry.”

II. A revision of the Introduction, in which we now state that our analyses on psychopathology should be regarded as exploratory, and require further testing in larger samples with more targeted hypotheses (p.4):

“Finally, in a series of exploratory analyses, we determined whether and how anxious individuals express a preference for punishment avoidance goals. In so doing, we found preliminary evidence that the degree of reliance on a goal-perseveration strategy to avoid punishment was positively associated with dispositional worry, which appears unique to those expressing worry and not to individuals with obsessive-compulsive or somatic anxiety symptoms.”

III. A revision of the Discussion to emphasize the tentative nature of conclusions we can draw regarding our analyses on worry, and also to consider that relationships between symptoms and cognitive indices may differ in a clinical population (p.18):

“Given that present results are preliminary in nature, future studies will need to test a prediction that chronic worry is associated with punishment perseveration in a larger sample. This should also include testing whether this association holds in a clinical population, as variation in symptoms in a clinical population may relate to punishment perseveration differently (Imperiale et al., 2021; Groen et al., 2019).”

B. More comprehensive reporting of clinical analyses

To further address the reviewers’ concerns, we expanded the reporting of the clinical analyses as follows:

I. We now report the adjusted p-values using a family-wise error correction approach for exploratory research (Rubin, 2017), and we explicitly note that correction typically employed in confirmatory research would render the results insignificant. The cited exploratory approach defines the family of tests by the number of tests within a given exploratory hypothesis. In the present set of analyses, we consider two exploratory hypotheses: (1) that anxiety is associated with greater GP punishment and (2) that anxiety is associated with greater MB punishment. We multiplied α level by 2 for both hypotheses because we explored whether these processes were associated with either somatic anxiety or chronic worry, controlling for co-occurring psychopathology and traits of non-interest (IQ, attention and effort). Our approach, and its associated caveat, are now noted in Results (p.15):

“It is important to note that all aforementioned p-values testing our key hypotheses (Table 2B) are corrected for multiple comparisons using a correction procedure designed for exploratory research (Rubin, 2017), which only controls for number of statistical tests within each hypothesis. Using a more conservative Bonferroni error correction for all 4 regression models, as typically employed in hypothesis-driven confirmatory work (Frane, 2020), resulted in a p-value for the key effect of worry and punishment perseveration that no longer passed a conventional significance thresholds (p=0.08). Thus, future work with a more targeted, hypothesis-driven approach needs to be conducted to ensure our tentative inferences regarding worry are valid and robust.”

II. To make transparent the associations between key variables in the model and forms of psychopathology, we now present in the main text and in new Figure 5 all relevant bivariate relationships, including with somatic anxiety. We have thus added the following to Results (p.14):

“We first report the bivariate relations between each form of psychopathology and inverse temperature parameters reflecting tendencies to utilize MB and GP punishment avoidance. Given that individuals with OCD and anxiety symptoms may over-prioritize threat detection, it is conceivable that there is a relationship between all three forms of psychopathology and model-based punishment avoidance. However, we found no significant or trending relationships between any form of psychopathology and model-based control for punishment avoidance (Figure 5A, left column). An alternative possibility is that individuals with anxiety suffer from a dysregulation in goal pursuit, reflecting a failure to disengage punishment avoidance when instructed to do so. On this basis, we explored whether worry and somatic anxiety are positively associated with goal-perseveration for punishment avoidance. In so doing we found initial evidence of a trending relationship between the tendency to worry and punishment avoidance perseveration (B=2.15, t = 1.4, p=0.16; Figure 5A, right column).”

III. The reviewers correctly point out that individual parameters were non-independently estimated. However, these only serve as predictors in the reported regression analyses, and thus, these analyses make no assumption that they were independently sampled (Hastie et al., 2009). We now clarify this issue in Methods (p.21):

“Although the computational parameters were non-independently estimated by our hierarchical model-fitting procedure, it is vital to note this does not compromise the validity of the least-squares solution to the regressions we ran. Indeed, Friedman, Hastie, and Tibshirani (2009) show that, ‘Even if the independent variables were not drawn randomly, the criterion is still valid if the dependent variables are conditionally independent given the [independent variable] inputs’ (p.44).”

Nevertheless, to mitigate doubt regarding the results of our regression analyses, we conducted nonparametric permutation tests, wherein we shuffled the task data with respect to the psychopathology scores. We include these nonparametric analyses in Results (p.14):

“For each regression model, we computed p-values using a nonparametric permutation test wherein we shuffled the task data with respect to the psychopathology scores, repeating the analysis on each of 10,000 shuffled datasets to derive an empirical null distribution of the relevant t-statistics.”

IV. To motivate use of covariates in the regression analyses, we now expound in Results on why we included each covariate. Additionally, we now validate the inclusion of covariates by showing that a regression with the covariates predicts left-out data better than regression without covariates (p.14-15):

“To provide a more specific test of our key hypotheses, we removed variance of non-interest in order to sensitize our analyses to unique relationships between forms of psychopathology and types of punishment avoidance. Firstly, generalized, as opposed specific obsessive, worry is thought to be particularly associated with difficulty in disengaging from worry (Berenbaum, 2010), since it lasts significantly longer in both clinical (Dar and Iqbal, 2015) and community samples (Langlois, Freeston, and Ladouceur, 2000). Thus, we dissociated generalized from obsessive worry using the same approach taken in previous studies (Doron et al., 2013; Stein et al., 2010), namely, by including a measure of OCD symptoms as a control covariate. Controlling for OCD symptoms has the additional benefit of accounting for known relations between OCD and poor learning of task structure, reduced model-based control, and perseverative tendencies (Gillan et al., 2016; Seow et al., 2021; Sharp et al., 2021). Secondly, another potentially confounding relationship exists between worry and somatic anxiety (e.g., Sharp, Miller and Heller, 2015), likely reflecting a general anxiety factor. Thus, we isolated worry by controlling for somatic anxiety, as commonly done in studies seeking to quantify distinct relationships of worry and somatic anxiety with cognitive performance (e.g., Warren, Miller and Heller, 2021) or associated neural mechanisms (e.g., Silton et al., 2011). Finally, we controlled for covariance between computational strategies that might reflect general task competencies. This included the utilization of MB (including learning rates and inverse temperatures) since observed anticorrelations in the empirical data (Figure S7) between GP and MB may derive from causal factors such as attention or IQ, as well as a general tendency to mitigate cognitive effort by using less costly strategies (AP, MF and GP inverse temperatures; Figure S7). This analysis showed a stronger relationship between worry and punishment perseveration (ß=3.14 (1.38), t = 2.27, p=0.04, Figure 5C). No other significant relationship was observed between punishment perseveration or model-based punishment avoidance and psychopathology (Figure 5C). Ultimately, we validated the full model using a 5-fold cross-validation procedure which showed that regressing worry onto the aforementioned covariates (using a ridge regression implementation) explains significantly more variance in left out test-set data (R^2^=0.24) relative to the models of the bivariate relationships between worry and GP Punishment (R^2^=0.01) and MB Punishment (R^2^=0.00).”

2) With respect to the modelling of behaviour, (i) more information about parameter recovery is requested, and (ii) concerns were raised about parameter estimation more generally given the skew evident in the inverse temperatures, coupled with the multicollinearity. More information should be provided here, including proportion of recovered parameters that can capture the main parameter-based results of the paper.

We thank the reviewers for raising these concerns regarding parameter multicollinearity, skew, and recoverability. To address these, we now show that multicollinearity involving the key inverse temperature parameters is low, and clarify that skew is expected given the inherently skewed γ priors that we (and many others) use to model these parameters. Additionally, we demonstrate that these parameters are highly recoverable, in line with levels reported in extant computational modelling literature. The changes to the manuscript include:

I. Multicollinearity. We show that multicollinearity involving key parameters of interest is relatively low, within acceptable levels with respect to prior studies. To report levels of multicollinearity more comprehensively, we now include a full heatmap of correlations between fitted parameters in Figure 4 —figure supplement 4. The only large correlation was between the learning rate for factual and counterfactual feature outcomes (r=0.68). Modest correlations were also observed between model-based inverse temperatures (r=0.37) and between model-based change parameters (r=0.34).

Moreover, we clarify the levels of multicollinearity between fitted parameters in Methods (p.21):

“We report all bivariate correlations between fitted parameters in Figure 4 – Supplementary figure 4.”

II. Skew. We clarify that skew is expected for key inverse temperature parameters, given that they were modelled with γ prior distributions that are inherently skewed. Indeed, extant studies show similar levels of skew in parameter distributions modelled using γ priors (Gillan et al., 2016; Sharp, Dolan and Eldar, 2021). We now clarify this issue in Methods (p.25):

“We note that the skew in inverse temperature parameters is to be expected given their Γ prior distributions are inherently skewed (Gillan et al., 2016; Sharp, Dolan and Eldar, 2021).”

III. Recoverability. As the Reviewers point out, an inability to dissociate between certain parameters is a common problem in reinforcement learning modelling (e.g., Palminteri et al., 2017). Thus, we now provide recoverability levels for all model parameters and how they trade-off against each other, in new Figure S4. Recoverability of the four parameters of interest, MB and GP for reward and punishment, spans the correlation range of 0.76 to 0.91, as consistent with levels of parameter recovery in extant studies (Haines, Vasilleva and Ahn, 2018; Palminteri et al., 2017). Additionally, all between-parameter (i.e., off-diagonal) correlations involving the key model parameters were low (i.e., weaker than 0.16), showing that the experimental design and model-fitting procedure were capable of successfully dissociating between these parameters (Wilson and Collins, 2019). This is now reported in Results (p.12):

“We validated that MB, GP and MF inverse temperature parameters were recoverable from simulated experimental data, and that the degree of recoverability (i.e., the correlations of true and recovered parameter values, which were between 0.76 and 0.91; Figure S4) was in line with extant reinforcement learning modelling studies (Haines, Vasilleva and Ahn, 2018; Palminteri et al., 2017). Similarly, low correlations between estimated parameters (all weaker than 0.16) demonstrate our experimental design and model-fitting procedure successfully dissociated between model parameters (Wilson and Collins, 2019).”

3) The interpretation of the key behavioural effects: could the authors defend the interpretation that GP is not an actual strategy, but rather a noisy or distracted attempt at MB learning. It was felt that it is rather a big claim to say that the GP approach is a reasonable third strategy on top of MF/MB learning and if this interpretation is to be maintained, the authors need to back this up by showing it is something more than just an imprecise MB approach.

We thank the reviewers for an opportunity to both clarify our interpretation of how GP relates to MB and MF, as well as expand our argument that GP is a strategic heuristic, and not simply a noisy or distracted attempt at MB learning. Below, we begin by clarifying that, since it shares its data structures with MB, GP is best thought of as a heuristic alteration to MB learning. We then argue that this heuristic is indeed a resource rational strategy, not simply a goal-forgetting MB agent.

Firstly, a purely forgetful agent would not demonstrate the observed valence effect that is rational for agents wishing to avoid potentially fatal punishment. Secondly, our experiment was specifically designed to prevent forgetting in that the instructed goal was presented on the screen for the entire duration that participants deliberated their decisions. Thirdly, we show through formalizing a model of MB forgetting that a key prediction of that model – namely, that GP and MB utilization should be highly positively correlated – does not hold in our empirical data. Finally, we expound on the various ways that GP computations save costly resources while producing high-performance (in terms of reward earned) policies.

I. The GP system is a strategically modified model-based strategy.

We first clarify that GP corresponds to a MB agent that strategically avoids goal-switching, and therefore is not as distinct from MB as might have been suggested in the previous version of our manuscript. To emphasize this in the text, we have added the following to Results (p.6):

“An alternative strategy, that we term “goal-perseveration” (GP), might strike a better balance between simplicity and effectiveness. This strategy inherits the exact same knowledge of feature probabilities acquired by model-based learning, but simplifies action selection by persistently avoiding punishment and seeking reward, simultaneously, regardless of instructed goal. This, in principle, eliminates effortful goal-switching while utilizing all available information about the changing action-feature mapping. Thus, rather than constituting a separate decision system in its own right, GP is best thought of as a behavior produced by a strategic cost-saving MB agent.”

II. Forgetting the reward function would not predict a valence effect in GP.

A non-strategic forgetful MB agent would be just as likely to pursue the wrong goal regardless of which goal is presented on screen. This would be inconsistent with the significant valence effect that we observed, where participants tended to pursue the uninstructed goal predominantly during reward trials. The observed valence effect is consistent, instead, with a rational strategy proposed in prior work (Woody and Szechtman, 2011). Specifically, a punishment avoidance system should be far more attuned to false negatives (failing to detect a true threat) than its reward system counterpart because such missed attempts to avoid punishment could be fatal. We now highlight this point further in the Discussion (p.16):

“The strategic deployment of GP primarily towards punishment avoidance indicates such behavior is not merely a reflection of a noisy or forgetful MB system. Indeed, our finding that humans use less flexible computational strategies to avoid punishment, than to seek reward, aligns with the idea of distinct neural mechanisms supporting avoidance and approach behavior (McNaughton and Gray, 2000; Lang, Bradley and Cuthbert, 1998). Moreover, comparative ethology and evolutionary psychology (Pinker, 1997) suggest there are good reasons why punishment avoidance might be less flexible than reward seeking. Woody and Szechtman (2011) opined that “to reduce the potentially deadly occurrence of false negative errors (failure to prepare for upcoming danger), it is adaptive for the system to tolerate a high rate of false positive errors (false alarms).” Indeed, we demonstrate that in the presence of multiple shifting goals, perseverance in punishment avoidance results in false positives during reward seeking (Figure 3B), but avoids ‘missing’ punishment avoidance opportunities because of lapses in goal switching (Figure 3C). Future work could further test these ideas, as well as potential alternative explanations (Dayan and Huys, 2009).”

III. A model-based strategy that forgets the reward provides a worse account of participant choices than a combination of MB and GP strategies.

Reviewer 2 (comment 2.1) additionally suggested that a forgetful MB system could potentially explain the valanced effect of GP if forgetting of the goal set occurred more often on either reward or punishment trials. Although we argue that a dependence of forgetting on goal valence suggests that such forgetting would itself be strategic, in the interest of determining whether such an account could in principle explain our results, we now fit a new model (Forgetful + MB + MF + AP) that addresses this potential mechanism. This model included all components of the winning model, except for the GP system, and extended the model-based system to include two additional parameters, fR and fp which determine the probability of forgetting the presented reward vector (which defines the goal) on reward trials and punishment trials respectively. Specifically, according to the model, on each trial according to a fixed probability (either fR and fp depending on whether the trial had a reward or punishment goal) the participant replaced the instructed goal with the opposite goal (e.g., if the actual goal was punishment avoidance, the participant used the reward pursuit goal).

We found that this model fit the data worse than a model which included separate MB and GP controllers with no goal forgetting (MB + MF + GP + AP), thereby confirming that a model where the model-based controller forgets the current goal with different rates on reward and punishment trials does not account for our results supporting GP as well. We have added this modelling result in Figure 4 —figure supplement 3 and amended the figure to include the new BIC for this model.

In order to investigate what aspects of the data were better captured by the winning model, we used the forgetful model (Forgetful + MB + MF + AP) to simulate a new dataset, using the best-fit parameters for each participant. We predicted that because in this model both MB and (apparent) GP-like effects on choice emerge from a single MB strategy (that at times forgets the rewards) then such effects should be correlated in the simulated dataset. We confirmed this prediction by fitting the GP model (MB + MF + GP + AP) to this simulated data, and showing that MB and GP parameters, within a valence, were indeed correlated. In contrast, when fit to the actual data, no correlation between MB and GP effects are detected, suggesting that these do not emerge from a single MB system, but instead reflect distinct task strategies. We present Author response image 1 the empirical GP correlations from our data on the left, and the simulated GP correlations generated by forgetful MB agents:

**Author response image 1. sa2fig1:** A comparison of GP-MB correlations for empirical data (left) and for data simulated using forgetful-MB agents (right).

We have added this text to our model comparison results detailed in Figure 4 —figure supplement 4:

“Finally, we tested an alternative model where GP behavior may derive from a MB strategy that occasionally forgets the reward function (Forgetful-MB+MthF+AP), allowing this forgetting to occur at different rates during reward and punishment goals. This model-based agent includes two additional parameters, fR and fp, which govern the probability of forgetting the presented reward function on reward pursuit trials and punishment avoidance trials respectively. Thus, on each trial, the model replaces the instructed goal with the opposite goal (e.g. if the actual goal was [-1, 0], the participant used [0,1]) with some fixed probability (either fR and fp, depending on the trial type). We again found that this model fit worse than the winning model, confirming that a model where the model-based controller forgets the current reward with different rates on reward and punishment trials does not account for our results supporting GP as well.”

IV. Availability of reward information on screen makes forgetting of reward information implausible. In addition to the aforementioned empirical evidence for why a forgetful MB agent is unlikely to explain our results, we also clarify here that we designed the task so as to reduce this specific kind of forgetting. We did so by continuously presenting the instructed goal on the screen while participants made their choice (presented in Author response image 2). The instructed goal only disappeared once a choice was made.

**Author response image 2. sa2fig2:** A depiction of what participants saw on the screen for the entire decision period in our multigoal reinforcement learning task.

Thus, any forgetting could be easily remedied by glancing at the screen. We now highlight in Results (p. 5) this additional reason to discount the possibility the participants forgot the current reward instructions:

“Note that the reward value of either feature was continuously presented throughout the choice deliberation time (Figure 1b), ensuring that there should be no reason for participants to forget the present trial’s goal.”

V. GP is a resource rational task strategy.

A necessary criterion for considering GP a heuristic strategy, as opposed to simply reflecting noise or error, is that it needs to fulfil a function that would cause it to be selected. This stands in contrast to errors, which do not fulfil such a function, and an agent would be better off avoiding. We suggest that GP fulfils a function of bypassing computational costs that make model-based RL computationally demanding, while still achieving good performance in the task. Our perspective here is inspired by the resource rationality framework, which argue that many heuristic deviations from seemingly rational strategies (in this task model-based reinforcement learning), rather than reflecting errors, instead reflect strategic balancing of task performance with computational costs of implementation (Gigerenzer and Goldstein, 2011; Lieder and Griffiths, 2020).

To see how GP may constitute a resource rational strategy, consider that the key feature of this task which makes model-based RL difficult to utilize is that the goals switch between trials. When the goals switch, the model-based agent is required to change which feature predictions it uses to derive action values, which takes time. At the same time, feature predictions need to be constantly updated even on trials for which a feature was not relevant, creating a burdensome dissociation between information used for decision and information used for learning. Like model-free RL, a GP agent entirely avoids these goal-switching costs, while obtaining substantially greater rewards than a model-free agent would. It does so by constantly using the same model-based feature predictions. Additionally, as noted above with regards to the observed valence effect, a GP agent that prioritizes avoiding losses can achieve this as well as model-based learning, while avoiding the relevant switching costs.

We have now elaborated in the Discussion (p. 14-15) how GP avoids the costs of MB control, and how it can constitute a resource rational heuristic that approximates model-based evaluation:

“GP may thus in fact constitute a resource-rational strategy (Lieder and Griffiths, 2020) for approximating MB control. To illustrate this, consider that model-based learning is computationally demanding in our task specifically because goals switch between trials. When the goals switch, a model-based agent must retrieve and use predictions concerning a different feature. Additionally, the agent needs to continuously update its predictions concerning features even when they are not presently relevant for planning. GP avoids these computationally costly operations by pursuing goals persistently, thus avoiding switching and ensuring that features are equally relevant for planning and learning. In this way, GP saves substantial computational resources compared to MB yet is able to perform relatively well on the task, achieving better performance than MF. Additionally, if a participant selectively cares about avoiding losses (for instance, due to loss aversion), GP can perform as well as MB. Thus, we propose the GP heuristic reflects a strategic choice, which can achieve good performance while avoiding the substantial resource requirements associated with model-based control. In this sense it fulfils a similar role as other proposed approximations to model-based evaluation including model-free RL (Sutton and Barto, 2018), the successor representation (Dayan, 1993; Momennejad et al., 2017), mixing model-based and model-free evaluation (Keramati, Smittenaar, Dolan, and Dayan, 2016), habitual goal selection (Cushman and Morris, 2015) and other identified heuristics in action evaluation (Daw and Dayan, 2014).”

This ties in with additional concerns about the exclusion of participants that did not obey certain rules of the task. It was not clear how their exclusion was justified given a valid interpretation of some of these key effects may be that (i) people find it hard to keep instructions in mind in complex tasks, (ii) people may be utilising strategies that you have not defined and are not well understood but are nonetheless real – e.g. belief that reward probabilities / 'luck' switches from trial to trial. Other concerns regarding exclusion were that they appear to be asymmetric with respect to reward/punishment conditions, suggesting these data are meaningful.

We thank the reviewers for pointing out the need to explain the rationale underlying our criteria for excluding participants. We first note that it is the norm in studies of learning and decision making to exclude participants whose performance is indistinguishable from pure guessing (e.g., Bornstein and Daw, 2013; Otto et al., 2013). Equivalently, in the present study our approach was to exclude only participants whose strategy was tantamount to performing at chance-level or below. We now show this by simulating each of the excluded strategies and measuring its performance. Furthermore, we measure the excluded participants’ actual accuracy and show that it was indeed indistinguishable from chance. In addition, we clarify that these participants’ fundamentally different model of the task makes it impossible to estimate the effects of interest for these participants. Finally, we conduct a sensitivity analysis in which we include these excluded participants and demonstrate that the key effects in our paper all hold.

I. Chance-level performance or worse among excluded participants and their strategies

We first verified through simulation that the strategies that we used as exclusion criteria yield equal or worse reward than purely guessing. Thus, we simulated the performance in our task of agents that either (i) treated reward features as punishment features, (ii) treated punishment features as reward features, or (iii) both (i.e., reversed the meaning of feature types), and compared them to agents that (iv) purely guess, (v) use both feature types in a model-based way, and finally (vi) use both feature types as intended but using a purely goal-perseveration strategy. We thus show that the strategies of participants we excluded (i, ii, and iii) do as poorly or worse than purely guessing. This motivation for the exclusion criteria is now reported in Methods (p.19):

“These errors in following task structure are fundamental failures that result in average performance that is as poor or worse than an agent that purely guesses which action to take at each trial (Figure 6).”

Secondly, we examined participants’ actual accuracy, by comparing their choices to those of an ideal observer model would make given the participants’ observations. In the excluded sample, participants on average chose correctly 49.65% of the time, whereas in the included group participants chose correctly 63.5% of the time (difference between groups: t=9.66,p<0.00001). We now add the following text to the Methods (p.19):

“Excluded subjects performed significantly worse in terms of choice accuracy. To derive accuracy, we computed the percentage of choices subjects made in line with an ideal observer that experienced the same outcome history (in terms of features) as each participant. On average, excluded subjects chose correctly 49.6% of time, whereas included subjects chose correctly 63.5% of time (difference between groups: t(190)=9.66, p<0.00001).“

II. Including subjects that misunderstand feature types would add noise to the hypothesis tests

Including participants we previously excluded would inject considerable noise into our estimation of how reward vs. punishment feature types were differentially utilized. For example, estimates of how individuals avoided punishment features would include participants that treated these features as if they were reward. Therefore, statistical tests of MB and GP goal valence differences would be contaminated by such subjects. In fact, testing whether punishment and reward goals were pursued to different degrees with a given strategy (which is the hypothesis of the study) is a non-sensical test for subjects that treated the task as if there was only one goal or that confused reward and punishment. Furthermore, given their distorted model of the task (where the task was reduced to one goal), these subjects had no incentive to use a more complex strategy than simple model-free learning, and thus their inclusion corrupts our ability to infer how subjects recruit different goal-directed systems for decision making. We now expand on these reasons in Methods (p.19):

“Additionally, including such subjects would reduce our sensitivity to estimating differences in the utilization of GP and MB for goals of differing valence, as such subjects treated the task as if there was only a single goal, or that the goals were opposite to their instructed nature. Moreover, given their model of the task, such subjects could approach the task optimally using a MF strategy, and thus would not be incentivized to use goal-directed strategies at all.”

III. Sensitivity Analysis

To mitigate any remaining concern about subject exclusion, we conducted a sensitivity analysis that aligns with Reviewer 3’s suggestion to determine if our key effects hold in the larger sample that includes subjects that incorrectly treat certain feature types as if they were another type. Modelling the large sample (n=242; 98% retention) required us to allow negative inverse temperature parameters, since these are required to account for subjects who for instance treated a reward feature as if it were a punishment feature and the converse. We thus replaced the γ priors for the inverse temperature parameters with normal distributions. The analyses for this larger sample resulted in all key results in the main paper holding, and in some cases, strengthening. We now note in the Methods (p.20):

“To determine whether our relatively strict subject exclusion policy might have affected the results, we conducted a sensitivity analysis on a larger sample (n=248; 98% retention) including subjects that mistreated the instructed value of certain features. To account for these subjects’ behavior, we used normal priors to allow negative inverse temperature parameters. Fitting these revised models to our data, we again demonstrate that our winning model was the best-fitting model compared to all other models. Second, we show that the GP valence effect held and even came out stronger in this larger sample. Thus, the mean difference in GP utilization for punishment and reward goals was 0.24 in our original sample and 0.50 in the larger sample (p< 0.0001). Finally, we show the MB valence effect also held in this larger sample (original sample mean difference between MB reward and MB punishment = 2.10 ; larger sample mean difference = 1.27, both p-values < 0.0001).”

Reviewer #1 (Recommendations for the authors):1. Psychopathology analyses.a. If the authors wish to make a connection to psychopathology, reporting the relationship between worry alone – rather than controlling for other overlapping symptom measures and model parameters – would be more appropriate. The recommendations for testing multiple, partially overlapping psychopathology measures in this paper may be helpful: DOI: 10.1177/21677026211017834

We agree with the reviewer that presenting bivariate relationships between psychopathology and task strategies is important and now present in full bivariate relations between all dimensions of psychopathology and the computational parameters of interest, which we detail in response to Editor comment 1, Points II and IV. However, as noted above, we have principled reasons for controlling for co-occurring forms of psychopathology, and thus have chosen to present both analyses.

b. An alternate approach would be to focus this paper on the main findings about learning strategies and to save relationships to psychopathology for a future paper with a more appropriate sample.

We thank the reviewer for the suggestion, which is indicative of the tentative nature of psychopathology findings which we agree are preliminary in nature. However, we believe a more transparent alternative is to include these analyses and re-frame them as exploratory, per R3’s suggestion. Doing so will enable future studies to target worry and goal-perseveration a priori. We thus amended our framing of the psychopathology analyses as exploratory in the Abstract, Introduction and Discussion. Each of the changes to the main text is detailed in response to Editor Comment 1, Re-framing clinical analyses and exploratory and preliminary.

2. Parameter-based analyses.a. Providing more information on parameter recovery is needed. In particular, showing the proportion of recovered parameters that can capture the main parameter-based results of the paper (Figure 2C/D) would show that these findings reflect true underlying parameter differences rather than artifacts of model estimation.b. If the authors retain the psychopathology analyses, they should be conducted in a way that does not assume independence of parameter estimates.c. Alternatively, the analyses using relative model fits and trialwise regressions provide most of the information needed for the conclusions of the paper. The parameter-based analyses could be omitted with the focus instead on these other kinds of analyses.

Although we agree with the reviewer that there is substantial overlap in the model-agnostic regression and parameter-based analyses, we opt for retaining both sets of analyses because only the computational modelling explains subjects’ choices beyond what can be explained solely based on the previous trial’s observations. Thus, we now emphasize the added value of analyzing the fitted parameters from the computational model in Results (p.11):

“The presence of unique signatures of MB, MF, and GP decision strategies in the empirical data presents strong evidence for the use of these strategies, but the signature measures are limited to examining goal-switch trials and, within those trials, examining the impact of features observed on the very last trial. To comprehensively quantify the extent to which participants utilized each strategy for reward seeking and punishment avoidance, we next developed a series of computational models that aim to explain all participant choices given the features observed on all preceding trials.”

Reviewer #2 (Recommendations for the authors):The authors used the term "predicted" quite a bit to describe associations. I don't think this is justified (they haven't really done any predictive analyses).

We apologize for the excessive use of causal language to describe regression results. We have now changed this terminology to “positively/negatively associated with” throughout the article.

If I understand correctly, the same 4 random walks were used for all participants (randomised between the 4 associations). Of the two shown, one looks much more stable than the other. It would be useful to see all 4 walks to see how comparable they are (if I am correct that the same 4 are used for all participants). If the walks are very different, should their allocation to the associations be controlled for in the analysis?

To clarify, the task included two types of random walks: the first was more volatile (the best bandit switched once per block), while the second had more irreducible uncertainty (significantly closer across both random walks to 0.5 probability), both of which make learning more difficult. Importantly, random walks were counterbalanced across subjects: in version 1 of the task, the reward feature took the first type of random walk (i.e., more volatility) and the punishment feature took the second type of random walk (i.e., more irreducible uncertainty). In task version 2, the feature:random walk mapping was flipped.

**Author response image 3. sa2fig3:** Random walks from task version 1. Here, the reward feature took a more volatile walk, whereas the punishment feature had greater irreducible uncertainty. In task version 2 (given to the other half of participants) the feature:random walk mapping was flipped.

To test whether there were significant differences in GP and MB valence effects as a function of which feature type was paired with a given random walk, we fitted hierarchical logistic regressions quantifying the reported model-agnostic signatures (Figure 2C in text) twice, so as to independently analyze data from each task version (i.e., using each feature:random walk mapping). This allowed us to compare whether GP and MB valence effects change as a function of the feature:random walk mapping. We demonstrate across both task versions that the signature of MB Reward > MB Punishment and the signature of GP Punishment > GP Reward. Moreover, there is striking consistency in estimates across task versions and substantial overlap in the estimated HDIs for each effect (Table S1; ‘HDI’ refers to the 94% highest density interval of the posterior, bounded by the 3% and 97% quantiles). We now report this new validatory analysis in Methods (p.25):

“We ensured there were no significant differences in the direction and significance of key effects across task versions by separately fitting our Bayesian logistic regression noted above to the subset of subjects that performed each task version. Doing so showed that all effects held and to a remarkably similar degree in both task versions (see full results in Supplemental Table 1).”

It would be useful to report the relationship between worry and the block effect (i.e. you suggest high worry is associated with higher GP/lower MB for losses-do worried people adapt to changes in the base rates of the outcomes?).

We thank the reviewer for this suggestion. We have now tested for a possibility of an interaction between worry and a block effect, and the results did not support this interaction. This analysis is now reported in Results (p.15):

“Of note, we additionally found no association between the parameter governing how MB punishment was modulated by task block and levels of worry, both when including worry alone (ß=2.5 (1.91), t=1.31, p=0.19) and when controlling for the same covariates as detailed above (ß=1.46 (1.65), t=0.88, p=0.38).”

Reviewer #3 (Recommendations for the authors):Well done on an interesting read and a contribution that will be informative for a lot of researchers. I have some suggestions to improve the paper.All analyses with the 3 clinical factors should be presented in full; including supplementary figures if possible. Simple associations should be carried out before adding covariates to assist the reader in interpreting these findings and in generating hypotheses based on them. OCD is said to be not related to parameters at p=.08, while worry is at p=0.04 (uncorrected i guess more like p=0.02 for the latter), these are not likely to be different from one-another. And they may depend on the inclusion of these variables in the same model. Reader needs more transparency around these effects and any claims of specificity need more support. The data presented actually suggests the opposite.

We thank the reviewer for helpful suggestions to improve the clarity and transparency of our clinical analyses. We now present in full all our analyses, as detailed in response to Editor comment 1. Of note here, the trending *negative* relationship between OCD and punishment perseveration was in the opposite direction of the relationship between punishment perseveration and worry. This is now clearly highlighted in an updated Figure 5**.**

Relatedly, the result in relation to worry, the effect is marginal at p=.04. While 2 multiple comparisons are controlled for, this is a fairly liberal decision given several tests were conducted and reported (i.e. GP MB and MF for punishment/reward = 6 at least; plus the 3 clinical scales = 18 etc). I'd encourage the authors to report all of the associations in a table, correct for multiple comparisons. This will serve the same purpose of suggesting the most interesting avenue for future research but also give the reader a fuller view on specificity of this to worry. This exploratory framing for the clinical effects does not detract from the main contribution of the paper or the potential for this to be especially interesting for 'worry' – it would just make them clearer and let the reader decide that for themselves a bit more.

We have changed the language about our hypotheses at the article’s outset, present all results in full, and present corrected and uncorrected p-values to be transparent about our correction for multiple comparisons. We temper our claims about the relation between worry and punishment perseveration in the Abstract, Introduction and Discussion, as detailed above in response to Editor comment 1, Re-framing clinical analyses and exploratory and preliminary.

There needs to be a bit more done with respect to relating the clinical variables to the model parameters. I would have thought this would be best placed within the hierarchical model itself. Alternatively, I wonder if these is a point-estimate that could be generated that is more flexible and less dependent on the overall group effects and other parameter values.The authors describe issues with collinearity of the parameter values. Can a correlation matrix in the supplement be included that reports these (I think currently you can sort of see it based on simulated vs real data, but this is not the same as correlating real vs real across params).

We now report this heatmap in Figure 4 —figure supplement 4, detailed in response to Editor comment 2.

I strongly encourage all subjects are retained (though i feel less strongly about excluding those not completing enough trials, 90% even seems a bit harsh/wasteful of data). If not, then a clear justification for why the strategy or approach of these subjects is not an accurate reflection of potentially the decision making preferences of 22% of the population. More standard indicators of inattentive responding focus on RTs, overly rigid responding that renders modelling choice impossible. Not clear why these were not used here as they seem better justified indicators of inattentive subjects. At the risk of belabouring the point(!), defining these subjects as 'not understanding instructions' could be applied to many of the key findings of this paper (i.e. avoidance perseveration suggests they don't pay attention to the current goals etc). So I think this practice is not ideal.

We agree with the reviewer that more comprehensive justification for, and scrutinization of, our exclusion criteria is warranted. We first demonstrate, by simulating agents that would be excluded in our study due to misunderstanding feature types, that the excluded strategies' performance in terms of points won is equal to, or worse than, that of agents that are purely guessing which action to take at each trial. By constrast, simulating a goal-perseveration strategy shows that it is far more adaptive in terms of reward earned and reduction in computation costs. We additionally show that excluded subjects as a group indeed perform at chance level on average. Finally, we show in a sensitivity analysis that if we include such subjects, all the major effects hold, and in some cases, become even stronger. We address this concern thoroughly in response to Editor comment 4.

References

1. Bornstein, A. M., and Daw, N. D. (2013). Cortical and hippocampal correlates of deliberation during model-based decisions for rewards in humans. PLoS Computational Biology, 9(12), e1003387.<milestone-start />‏<milestone-end />

2. Cushman, F., and Morris, A. (2015). Habitual control of goal selection in humans. Proceedings of the National Academy of Sciences, 112(45), 13817-13822.<milestone-start />‏<milestone-end />

3. Dar, K. A., and Iqbal, N. (2015). Worry and rumination in generalized anxiety disorder and obsessive compulsive disorder. The Journal of psychology, 149(8), 866-880.<milestone-start />‏<milestone-end />

4. Daw, N. D., and Dayan, P. (2014). The algorithmic anatomy of model-based evaluation. Philosophical Transactions of the Royal Society B: Biological Sciences, 369(1655), 20130478.<milestone-start />‏<milestone-end />

5. Dayan, P. (1993). Improving generalization for temporal difference learning: The successor representation. Neural Computation, 5(4), 613-624.<milestone-start />‏<milestone-end />

6. Doron, G., Derby, D. S., Szepsenwol, O., and Talmor, D. (2012). Tainted love: Exploring relationship-centered obsessive compulsive symptoms in two non-clinical cohorts. Journal of Obsessive-Compulsive and Related Disorders, 1(1), 16-24.<milestone-start />‏<milestone-end />

7. Frane, A. V. (2020). Misguided opposition to multiplicity adjustment remains a problem. Journal of Modern Applied Statistical Methods, 18(2), 28.<milestone-start />‏<milestone-end />

8. Friedman, J., Hastie, T., and Tibshirani, R. (2009). The elements of statistical learning (Vol. 1, No. 10). New York: Springer series in statistics.

9. Gigerenzer, G., and Goldstein, D. G. (2011). The recognition heuristic: A decade of research. Judgment and Decision Making, 6(1), 100-121.<milestone-start />‏<milestone-end />

10. Gillan, C. M., Kosinski, M., Whelan, R., Phelps, E. A., and Daw, N. D. (2016). Characterizing a psychiatric symptom dimension related to deficits in goal-directed control. eLife, 5, e11305.<milestone-start />‏<milestone-end />

11. Groen, R. N., Wichers, M., Wigman, J. T., and Hartman, C. A. (2019). Specificity of psychopathology across levels of severity: a transdiagnostic network analysis. Scientific reports, 9(1), 1-10.

12. Haines, N., Vassileva, J., and Ahn, WY (2018). The outcome ‐ representation learning model: A novel reinforcement learning model of the iowa gambling task. Cognitive Science , 42 (8), 2534-2561.

13. Haines, N., Kvam, P. D., Irving, L. H., Smith, C., Beauchaine, T. P., Pitt, M. A., and Turner, B. (2020). Theoretically Informed Generative Models Can Advance the Psychological and Brain Sciences: Lessons from the Reliability Paradox. PsyArxiv.

14. Imperiale, M. N., Lieb, R., Calkins, M. E., and Meinlschmidt, G. (2021). Multimorbidity networks of mental disorder symptom domains across psychopathology severity levels in community youth. Journal of psychiatric research, 141, 267-275.

15. Keramati, M., Smittenaar, P., Dolan, R. J., and Dayan, P. (2016). Adaptive integration of habits into depth-limited planning defines a habitual-goal–directed spectrum. Proceedings of the National Academy of Sciences, 113(45), 12868-12873.

16. Langlois, F., Freeston, M. H., and Ladouceur, R. (2000). Differences and similarities between obsessive intrusive thoughts and worry in a non-clinical population: Study 1. Behaviour Research and Therapy, 38(2), 157-173.<milestone-start />‏<milestone-end />

17. Leeuwenberg, A. M., van Smeden, M., Langendijk, J. A., van der Schaaf, A., Mauer, M. E., Moons, K. G., and Schuit, E. (2021). Comparing methods addressing multi-collinearity when developing prediction models. arXiv preprint arXiv:2101.01603.<milestone-start />‏<milestone-end />

18. Lieder, F., and Griffiths, TL (2020). Resource-rational analysis: Understanding human cognition as the optimal use of limited computational resources. Behavioral and Brain Sciences , 43.

19. Momennejad, I., Russek, EM, Cheong, JH, Botvinick, MM, Daw, ND, and Gershman, SJ (2017). The successor representation in human reinforcement learning. Nature Human Behavior , 1 (9), 680-692.

20. Otto, AR, Raio, CM, Chiang, A., Phelps, EA, and Daw, ND (2013). Working-memory capacity protects model-based learning from stress. Proceedings of the National Academy of Sciences , 110 (52), 20941-20946.

21. Palminteri, S., Lefebvre, G., Kilford, E. J., and Blakemore, S. J. (2017). Confirmation bias in human reinforcement learning: Evidence from counterfactual feedback processing. PLoS Computational Biology, 13(8), e1005684.<milestone-start />‏<milestone-end />

22. Rubin, M. (2017). Do p values lose their meaning in exploratory analyzes? It depends on how you define the familywise error rate. Review of General Psychology , 21 (3), 269-275.

23. Seow, T. X., Benoit, E., Dempsey, C., Jennings, M., Maxwell, A., O'Connell, R., and Gillan, C. M. (2021). Model-based planning deficits in compulsivity are linked to faulty neural representations of task structure. Journal of Neuroscience, 41(30), 6539-6550.<milestone-start />‏<milestone-end />

24. Sharp, PB, Dolan, RJ, and Eldar, E. (2021). Disrupted state transition learning as a computational marker of compulsivity. Psychological Medicine , 1-11.

25. Silton, R. L., Heller, W., Engels, A. S., Towers, D. N., Spielberg, J. M., Edgar, J. C., … and Miller, G. A. (2011). Depression and anxious apprehension distinguish frontocingulate cortical activity during top-down attentional control. Journal of Abnormal Psychology, 120(2), 272.<milestone-start />‏<milestone-end />

26. Stein, D. J., Fineberg, N. A., Bienvenu, O. J., Denys, D., Lochner, C., Nestadt, G., … and Phillips, K. A. (2010). Should OCD be classified as an anxiety disorder in DSM‐V?. Depression and anxiety, 27(6), 495-506.<milestone-start />‏<milestone-end />

27. Sutton, R. S., and Barto, A. G. (2018). Reinforcement learning: An introduction. MIT press.<milestone-start />‏<milestone-end />

28. Warren, S. L., Heller, W., and Miller, G. A. (2021). The structure of executive dysfunction in depression and anxiety. Journal of Affective Disorders, 279, 208-216.<milestone-start />‏<milestone-end />

29. Wilson, R. C., and Collins, A. G. (2019). Ten simple rules for the computational modeling of behavioral data. eLife, 8, e49547.<milestone-start />‏<milestone-end />

30. Woody, E. Z., and Szechtman, H. (2011). Adaptation to potential threat: the evolution, neurobiology, and psychopathology of the security motivation system. Neuroscience and Biobehavioral Reviews, 35(4), 1019-1033.<milestone-start />‏<milestone-end />